# ENHANCING DELTA COMPRESSION IN LLMS VIA SVD-BASED QUANTIZATION ERROR MINIMIZATION

## ABSTRACT

Fine-tuning is a crucial process for adapting large language models (LLMs) to diverse applications. In certain scenarios, like multi-tenant serving, a large number of LLMs finetuned from the same base model are deployed to meet complex requirements for users. Recent works explore delta-compression approaches to quantize and compress the delta weights between the customized LLM and the corresponding base model. However, they exhibit inadequate performance at high compression ratios due to their empirical nature. In this work, we introduce DELTAMIX, an adaptive mixed-precision delta-compression framework designed to minimize quantization error in the singular value decomposition (SVD) space without imposing additional assumptions. DELTAMIX provides a theoretical justification for the necessity of mixed-precision compression and presents a practical quantization solution that involves solving a 0/1 linear integer programming problem alongside a reconstruction target correction method. Experimental results across multiple models and benchmarks illustrate that DELTAMIX consistently outperforms all baseline methods. Notably, on tasks such as AIME2024 and GQA, DELTAMIX exceeds the performance of the best baseline, Delta-CoMe, by 22.3% and 6.1% for 7B parameter models, respectively.

## 1 INTRODUCTION

Large language models (LLMs) have shown breakthrough performance on various knowledge-intensive (Grattafiori et al., 2024; Team, 2024; Jiang et al., 2023) and complex reasoning tasks (DeepSeek-AI, 2025; Grattafiori et al., 2024). Enhancing deployment efficiency is crucial for facilitating LLM applications on edge devices and in cloud environments (Yao et al., 2024). In multi-tenant serving scenarios, multiple users fine-tune the same base model using their customized datasets (Wei et al., 2024; Yu et al., 2023), resulting in a variety of customized models that share a common foundation. These models, derived from the same base LLM (e.g., Qwen2.5 (Team, 2024) or LLaMA (Grattafiori et al., 2024)), need to be deployed concurrently to address simultaneous user requests. Conventional LLM compression approaches (Frantar et al., 2022; Lin et al., 2024) focus on quantizing and compressing the full model parameters. While effective at low compression ratios, these methods struggle to maintain model performance at high compression ratios, resulting in significant storage and computational overhead when deploying multiple customized LLMs.

In contrast to full model compression, delta-compression (Yao et al., 2024; Liu et al., 2024; Ping et al., 2024) decomposes a customized LLM into two components: the base model and the delta weights, which encapsulate the differences between the customized model and its corresponding base model. This approach emphasizes the compression of delta weights. Consequently, in multi-tenant environments, a single base model can be deployed alongside multiple sets of compressed delta parameters. Delta-compression achieves significantly higher compression rates than full model compression, thereby substantially reducing overall deployment costs. Researchers have explored effective approaches for delta-compression. Ryu et al. (2023) proposes a 1-bit quantization approach, termed BitDelta, to reduce the size of delta weights. Liu et al. (2024) leverages the low-rank characteristics of delta weights to improve storage efficiency through low-rank approximation. Delta-CoMe (Ping et al., 2024) introduces a mixed-precision delta-compression technique based on singular value decomposition (SVD), allocating higher-bit representations to singular vectors associated with larger singular values. Although these existing approaches demonstrate promising performance at

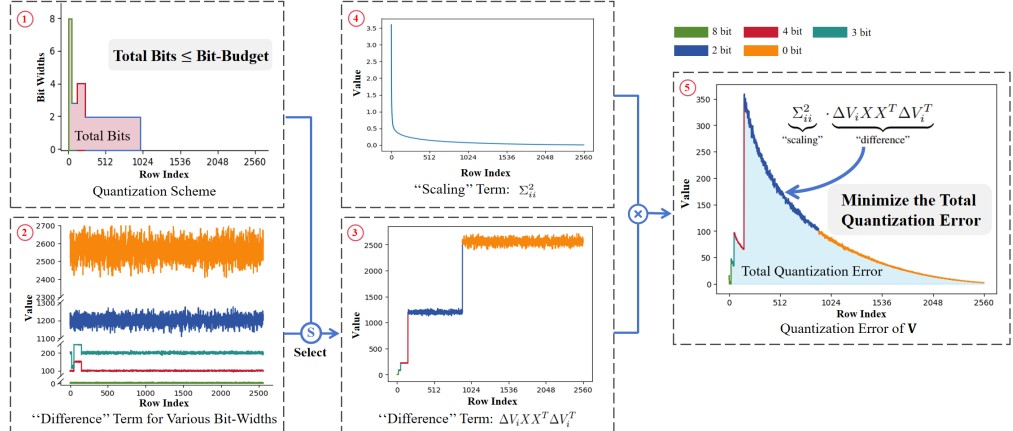

Figure 1: An overview of DELTAMIX. The quantization error of the $i$-th row of $\mathbf{V}$ comprise two components: a "scaling" term (④) and a "difference" term (③). The "scaling" term is fixed, while the "difference" term is related to the mixed-precision quantization scheme (①). DELTAMIX identifies the optimal quantization scheme within the constraints of the bit budget (①) to effectively balance these two components, thereby minimizing the total quantization error of $\mathbf{V}$ (⑤). Note that the "difference" term for various bit-widths (②) is pre-computed using a calibration dataset and remains fixed during the optimization process.

high compression ratios, they lack rigorous mathematical foundations, which can lead to suboptimal performance, especially in challenging compression scenarios.

In this work, we propose DELTAMIX, a high-performance mixed-precision delta-compression framework grounded in a solid theoretical foundation. DELTAMIX implements delta-compression within the SVD space, formulating the quantization objective as the minimization of layer-wise quantization error. By pursuing this objective, DELTAMIX establishes a mathematically sound mixed-precision compression strategy that accommodates flexible, user-defined compression ratios. This strategy derives **the mixed-precision quantization scheme** through the solution of a 0/1 linear integer programming problem and ensures optimization consistency throughout the quantization process via **a reconstruction target correction method**. Unlike Ping et al. (2024), which empirically posits that singular vectors corresponding to larger singular values are more significant and, therefore, necessitate higher-bit representations, DELTAMIX prioritizes the minimization of quantization error. It formulates all subsequent strategies based exclusively on this principle, eschewing reliance on singular values for assessing importance. This distinction is vital, as prior research has demonstrated that the significance attributed to singular values may not correlate with the performance of LLMs (Hsu et al., 2022; Wang et al., 2025).

We conduct extensive experiments on reasoning, math, code, and multimodal tasks across eight aligned LLMs to demonstrate the effectiveness of DELTAMIX. The results show that DELTAMIX achieves state-of-the-art performance among delta-compression methods, particularly in challenging scenarios where the norm of $\Delta \mathbf{W}$ is large. Notably, on the reasoning task AIME2024, DELTAMIX surpasses the leading baseline, Delta-CoMe, by 22.3% on the 7B model and 26.9% on the 14B model. Furthermore, DELTAMIX can achieve more than 6× GPU memory and disk storage savings, enabling the deployment of multiple models within constrained resource environments.

## 2 RELATED WORK

**Quantization Strategies for LLMs**  Quantization reduces the bit-precision of model parameters to lower GPU cost and accelerate inference. Current strategies for LLM quantization can be broadly categorized into quantization-aware training (QAT) and post-training quantization (PTQ). QAT simulates quantization operations during training and uses backpropagation to correct quantization errors (Zhou et al., 2018; Esser et al., 2020; Liu et al., 2023b; Wang et al., 2023). In contrast, PTQ quantizes a pre-trained model without further training, typically calibrating the quantized weights with a modest calibration dataset (Dettmers et al., 2022; Frantar et al., 2022; Lin et al., 2024; Lee et al., 2024). Given the high computational cost associated with training or fine-tuning large language

models, PTQ has become a particularly prevalent approach for LLM quantization. In our work, we leverage the GPTQ (Frantar et al., 2022) method within PTQ, focusing on mixed-precision quantization of the singular vectors of the delta parameters.

**Delta-Compression**  Delta-compression (Isik et al., 2023; Ryu et al., 2023; Liu et al., 2024; Ping et al., 2024) aims to diminish the storage and inference costs associated with serving multiple models by compressing delta parameters, which are the differences between the parameters of a fine-tuned LLM and its corresponding base LLM. GPT-Zip (Isik et al., 2023) extends GPTQ to compress the delta parameters into 2-bit, and then sparsify 95% of the quantized delta weights to further reduce storage costs. DeltaZip (Yao et al., 2024) extends the idea of structured pruning and delta-compression to develop a multi-tenant serving system. However, both methods are still limited to compression ratios of 2-bit and higher. Liu et al. (2024) introduces BitDelta, which compresses delta weight into 1-bit, using a trainable high-precision scaling factor for each delta weight matrix. From this point onward, the compression of delta parameters has entered the 1-bit era. In addition to these low-bit methods, Ryu et al. (2023) identifies the low-rank property of delta weights and achieves delta-compression through low-rank approximation. Recently, Delta-CoMe (Ping et al., 2024) leverages the benefits of both low-rank and low-bit compression methods, proposing a mixed-precision delta-compression method that uses varying bit-widths to represent different singular vectors of the delta weights. However, the rationale behind their mixed-precision quantization is predicated on a questionable hypothesis (Hsu et al., 2022; Wang et al., 2025): that singular vectors associated with larger singular values are inherently more important. This premise lacks a solid theoretical foundation, leading to a mixed-precision strategy that is primarily empirical and, consequently, suboptimal. In this work, we introduce DELTAMIX, which provides a mathematical proof of the necessity for mixed-precision in SVD-based delta-compression methods, and derives a quantization approach that is firmly grounded in mathematical theory.

## 3 METHOD

In this section, we introduce DELTAMIX, an adaptive mixed-precision delta-compression strategy for LLMs with mathematical support. In Section 3.1, we begin with the minimization of quantization error in the SVD space and derive the detailed quantization process. We provide a mathematical proof demonstrating the necessity of mixed-precision in this context. In Section 3.2, we introduce our mixed-precision schedule in detail, which is built on the solution of a 0/1 integer linear programming problem. Algorithm 1 shows the details of DELTAMIX.

---

**Algorithm 1** Algorithm for Quantization in DELTAMIX

**Data:** Delta parameter $\mathbf{W}$, List of candidate quantization bits $Q$, predefined averaged bit-width $G_b$, Calibration set $X$
**Result:** Quantized matrices $\hat{\mathbf{V}}$ and $\hat{\mathbf{U}}$
$\mathbf{U}, \mathbf{\Sigma}, \mathbf{V} \leftarrow \text{SVD}(\mathbf{W})$
**for** *bit b in Q* **do**
  $\mathbf{V_b} \leftarrow \text{SimQuant}(\mathbf{V}, b, X)$
  $\mathbb{E}_b^V \leftarrow \text{CalcLoss}(\mathbf{V}, \mathbf{V_b}, \mathbf{\Sigma})$
**end**
$B \leftarrow \text{CalcStorage}(Q)$
$S \leftarrow \text{SolveOpt}(B, G_b, \mathbb{E}^{\mathbb{V}})$
$\hat{\mathbf{V}} \leftarrow \text{QuantParams}(\mathbf{V}, \mathbf{S}, X)$
$\tilde{\mathbf{U}} \leftarrow \text{RTC}(\mathbf{U}, \hat{\mathbf{V}}, \mathbf{V}, \mathbf{\Sigma}, X)$
$\hat{\mathbf{U}} \leftarrow \text{QuantParams}(\tilde{\mathbf{U}}, \mathbf{S}, \hat{\mathbf{V}}, \mathbf{\Sigma}, X)$
**return** $\hat{\mathbf{V}}, \hat{\mathbf{U}}$;             // Return results

---

### 3.1 QUANTIZATION ERROR DERIVATION

At a high level, DELTAMIX follows the structure of the classical post-training quantization method GPTQ, by performing quantization to minimize the reconstruction error. Given a delta weight matrix $\mathbf{W}$ and the corresponding input $X$, the quantization objective of the GPTQ is to find a quantized matrix $\hat{\mathbf{W}}$ which minimizes the squared error:

$$\arg \min_{\hat{\mathbf{W}}} \left\| \mathbf{W}X - \hat{\mathbf{W}}X \right\|_F^2 = \sum_i \left\| W_i X - \hat{W}_i X \right\|_F^2 \approx \sum_i e_i \tag{1}$$

Following previous work (Hassibi et al., 1993; Nagel et al., 2020), the quantization error of the $i^{\text{th}}$ row of $\mathbf{W}$ can be approximated with a second-order Taylor expansion $e_i$:

$$e_i = \frac{1}{2} \Delta W_i \mathbf{H}_i \Delta W_i^T \tag{2}$$

Here $\Delta W_i = W_i - \hat{W}_i$ is the quantization difference of $i^{\text{th}}$ row, while the Hessian matrix $\mathbf{H}_i = 2XX^T$ is independent and identical across different rows in $\mathbf{W}$. By reusing $\mathbf{H}$, GPTQ derives the optimal quantized weights $\hat{\mathbf{W}}$ row by row, allowing for parallel computation across multiple rows.

Instead of directly quantizing $\mathbf{W}$, DELTAMIX performs quantization in the SVD space, by finding a quantized matrix $\hat{\mathbf{U}}$ and $\hat{\mathbf{V}}$ which minimizes the squared error:

$$\arg\min_{\hat{\mathbf{U}}, \hat{\mathbf{V}}} \left\| \mathbf{U}\mathbf{\Sigma}\mathbf{V}X - \hat{\mathbf{U}}\mathbf{\Sigma}\hat{\mathbf{V}}X \right\|_F^2 \tag{3}$$

where $\mathbf{W} = \mathbf{U}\mathbf{\Sigma}\mathbf{V}$. Below, we introduce the detailed quantization process of DELTAMIX, which first quantizes $\mathbf{V}$, and then moves to $\mathbf{U}$.

### 3.1.1 QUANTIZE $\mathbf{V}$

In this section, we present a theoretical analysis that motivates the need for mixed-precision quantization. Specifically, we find the quantized $\hat{\mathbf{V}}$ with the row-by-row approach by minimizing the squared error:

$$\arg\min_{\hat{\mathbf{V}}} \left\| \mathbf{U}\mathbf{\Sigma}\mathbf{V}X - \mathbf{U}\mathbf{\Sigma}\hat{\mathbf{V}}X \right\|_F^2 \approx \sum_i e_i^{\mathbf{V}}$$

$$e_i^{\mathbf{V}} = \frac{1}{2}\Delta V_i \mathbf{H}_i^{\mathbf{V}} \Delta V_i^T \tag{4}$$

Here $\Delta V_i = V_i - \hat{V}_i$ is the quantization difference of the $i^{\text{th}}$ row, and $\mathbf{H}_i^{\mathbf{V}} = 2\Sigma_{ii}^2 \cdot XX^T$ is the Hessian matrix of the $i^{\text{th}}$ row of $\mathbf{V}$ (with derivation details in Appendix C.1). As $\Sigma_{ii}^2$ is a scalar, we can reformulate the Eq. (4) as follows:

$$e_i^{\mathbf{V}} = \frac{1}{2}\Delta V_i \mathbf{H}_i^{\mathbf{V}} \Delta V_i^T = \underbrace{\Sigma_{ii}^2}_{\text{"scaling"}} \cdot \underbrace{\Delta V_i XX^T \Delta V_i^T}_{\text{"difference"}} \tag{5}$$

From Eq. (5), it is evident that the error for $i$-th row of $\mathbf{V}$ comprises two components: a "scaling" term $\Sigma_{ii}^2$, which suggests that rows (singular vectors) with larger singular values has larger scaling factor, and a "difference" term $\Delta V_i XX^T \Delta V_i^T$, derived from the quantization differences $\Delta V_i$ and limited sampling over a calibration set.

As illustrated in Figure 2, we present the results of the "scaling" and "difference" terms across different rows. The variation in the "difference" term remains relatively minor when the same bit-width is used to quantize different rows. In contrast, the "scaling" term decreases sharply as the row index $i$ increases. Consequently, the quantization error $e_i^{\mathbf{V}}$, which encompasses both terms, varies significantly across different rows under a uniform bit-width for quantization. To minimize the total error, it is ideal for the

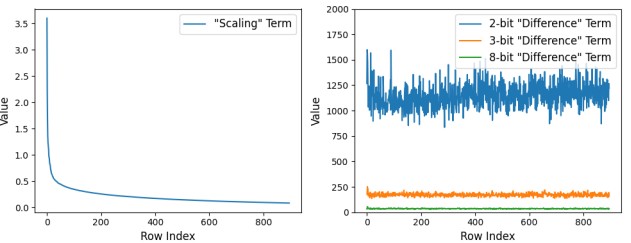

Figure 2: (Left) The value of "scaling" term (Eq. 5) at different row indices. (Right) The value of "difference" term ((Eq. 5) with different quantization bit-width at different row indices. We compute all results using Q_Proj at the last layer of Qwen2.5-Math-7B-Instruct.

quantization error of each row to be small. Given that the "scaling" term is fixed for each row, we can only adjust the "difference" term by carefully allocating bit-widths. However, due to the constraints of the total bit budget, we cannot allocate high bit-widths to all rows simultaneously. Therefore, we propose a strategy of assigning varying bit-widths to different rows to reduce the overall quantization error. **Eq. (5) provides a theoretical foundation for the necessity of mixed-precision quantization in SVD-based delta-compression**. We discuss the detailed mixed-precision schedule in Section 3.2, which allocates varying bit-widths to different rows, specifically the different singular vectors of $\mathbf{U}$, by formulating a 0/1 integer linear programming problem.

### 3.1.2 QUANTIZE $\mathbf{U}$

In this section, we analyze why mixed-precision quantization is not crucial for $\mathbf{U}$. After quantizing $\mathbf{V}$ to $\hat{\mathbf{V}}$, the quantization objective of $\mathbf{U}$ is:

$$
\arg\min_{\hat{\mathbf{U}}} \|\mathbf{U}\boldsymbol{\Sigma}\hat{\mathbf{V}}X - \hat{\mathbf{U}}\boldsymbol{\Sigma}\hat{\mathbf{V}}X\|_F^2 \approx \sum_i e_i^{\mathbf{U}}
$$

$$
e_i^{\mathbf{U}} = \frac{1}{2}\Delta U_i \mathbf{H}_i^{\mathbf{U}} \Delta U_i^T = \Delta U_i \boldsymbol{\Sigma}\hat{\mathbf{V}}X X^T \hat{\mathbf{V}}^{\mathbf{T}}\boldsymbol{\Sigma}^{\mathbf{T}}\Delta U_i^T
$$

(6)

Here $\Delta U_i = U_i - \hat{U}_i$, and the Hessian matrix of the $i^{\text{th}}$ row of $\mathbf{U}$ is given by $\mathbf{H}_i^{\mathbf{U}} = 2\boldsymbol{\Sigma}\hat{\mathbf{V}}X X^T \hat{\mathbf{V}}^{\mathbf{T}}\boldsymbol{\Sigma}^{\mathbf{T}}$(with derivation details in Appendix C.2). Upon comparing Eq. (5) and Eq. (6), we observe that $e_i^{\mathbf{U}}$ does not incorporate the scaling term present in Eq. (6). Consequently, when different rows are quantized using the same bit-width, there is no significant variation in error. This uniformity arises from the fact that the Hessian matrices for different rows of $\mathbf{U}$ are identical. Thus, unlike $\mathbf{V}$, there is no necessity to employ mixed precision when quantizing different rows of $\mathbf{U}$.

Therefore, DELTAMIX **determines the mixed-precision quantization schedule based on $\mathbf{V}$**, and then applies the same schedule to $\mathbf{U}$ for simplicity. Specifically, DELTAMIX quantizes $\mathbf{U}$ using a column-wise mixed-precision schedule, where the $i^{\text{th}}$ column of $\mathbf{U}$ adopts the same bit-width as the $i^{\text{th}}$ row of $\mathbf{V}$ as they correspond to the same singular value. Notably, DELTAMIX exhibits insensitivity to column-wise precision schedules, since GPTQ compensates for quantization-induced errors in the column direction by adjusting the unquantized weights during the quantization process. This compensation, however, does not occur between different rows, as different rows are independently quantized in GPTQ. This further underscores the importance of discussing row-wise mixed precision strategies aimed at minimizing the quantization error of $\mathbf{V}$. In Appendix E.1, we further demonstrate experimentally that applying the same mixed-precision quantization strategy to both $\mathbf{V}$ and $\mathbf{U}$ yields satisfactory performance.

**Reconstruction Target Correction.** In Eq. (6), we quantize $\mathbf{U}$ to reconstruct the target $\mathbf{U}\boldsymbol{\Sigma}\hat{\mathbf{V}}X$, which deviates from the initial target $\mathbf{U}\boldsymbol{\Sigma}\mathbf{V}X$. This deviation can negatively impact the performance of the quantized model. A straightforward approach to address this issue is to directly replace the reconstruction target with $\mathbf{U}\boldsymbol{\Sigma}\mathbf{V}X$; however, this would inhibit the application of GPTQ for quantization. Therefore, we propose a method termed "Reconstruction Target Correction" (RTC) to reduce the bias by transforming $\mathbf{U}\boldsymbol{\Sigma}\hat{\mathbf{V}}X$ in Eq. (6) to $\tilde{\mathbf{U}}\boldsymbol{\Sigma}\hat{\mathbf{V}}X$, where $\tilde{\mathbf{U}}$ is derived from the following equation:

$$
\min_{\tilde{\mathbf{U}}} \left\|\mathbf{U}\boldsymbol{\Sigma}\mathbf{V}X - \tilde{\mathbf{U}}\boldsymbol{\Sigma}\hat{\mathbf{V}}X\right\|_F^2
$$

$$
\Rightarrow \tilde{\mathbf{U}} = \mathbf{U}\boldsymbol{\Sigma}\mathbf{V}X X^T \hat{\mathbf{V}}^{\mathbf{T}}\boldsymbol{\Sigma}^{\mathbf{T}}(\boldsymbol{\Sigma}\hat{\mathbf{V}}X X^T \hat{\mathbf{V}}^{\mathbf{T}}\boldsymbol{\Sigma}^{\mathbf{T}})^{-1}
$$

(7)

See Appendix C.3 for detailed derivations. In summary, prior to quantizing $\mathbf{U}$, we first update $\mathbf{U}$ to $\tilde{\mathbf{U}}$ using Eq. (7). Subsequently, we perform quantization by minimizing $\|\tilde{\mathbf{U}}\boldsymbol{\Sigma}\hat{\mathbf{V}}X - \hat{\mathbf{U}}\boldsymbol{\Sigma}\hat{\mathbf{V}}X\|_F^2$. This approach aims to ensure that the reconstruction target closely approximates the original, without compromising the application of GPTQ for quantization.

### 3.2 OPTIMIZATION PROBLEM MODELING

In this section, we formulate the optimal mixed-precision bit allocation problem as a 0/1 integer linear programming model (see Eq. (8)). Given a user-specified compression target bit $G_b$, a candidate set of quantization bit-widths $Q$ of size $N_b$, and an upper bound $f_{\max}$ on the number of active bit-widths, the proposed model minimizes the quantization error by automatically selecting an subset of active bid-widths from $Q$, subject to the constraints imposed by $G_b$ and $f_{\max}$.

As shown in Eq. (8), the objective is to minimize the total quantization error, expressed as $\sum_i \mathbb{E}_i^{\mathbf{V}}\mathcal{S}_i^T$. Here, $\mathbb{E}_i^{\mathbf{V}} \in \mathbb{R}^{1 \times N_b}$ denotes the quantization error associated with different bit-widths for the $i^{\text{th}}$ row of $\mathbf{V}$, computed using predefined calibration data samples $X_n$ in accordance with Eq. (4). $\mathcal{S}_i \in \mathbb{R}^{1 \times N_b}$ is a binary optimization variable indicating the selected bit-width for quantizing the $i^{\text{th}}$ row of $\mathbf{V}$ and the corresponding $i^{\text{th}}$ column of $\tilde{\mathbf{U}}$. Note that our objective is limited to the quantization error of $\mathbf{V}$, with a detailed discussion provided in Sections 3.1.1 and 3.1.2.

$$\min_{\mathcal{S}} \sum_i \mathbb{E}_i^{\mathbf{V}} \mathcal{S}_i^T \qquad \text{(Total quantization error)}$$

$$\text{s.t.} \sum_i \mathcal{S}_i B \leq G_b(h_{\text{in}} \cdot h_{\text{out}}) \qquad \text{(Bit budget constraint)}$$

$$\text{sum}(S_i) = 1 \qquad \text{(One-hot vector constraint)}$$

$$S_i - f \leq 0 \qquad \text{(Bit-width selection constraint)}$$

$$\text{sum}(f) \leq f_{\max} \qquad \text{(Bit-width number constraint)}$$

(8)

The optimization problem has four constraints. (1) The "bit-budget constraint" ensures that the quantized model achieves a target compression bit that does not exceed the predefined threshold $G_b$. Here $h_{\text{in}}$ and $h_{\text{out}}$ represent the input and output dimension of $\mathbf{W}$. $B \in \mathbb{R}^{N_b \times 1}$ represents the storage required for quantizing a row of $\mathbf{V}$ and a column of $\tilde{\mathbf{U}}$ at different bit-widths, which is computed as $B = (h_{in} + h_{out}) \cdot Q$. (2) The "one-hot vector constraint" requires that each row of $\mathbf{V}$ and the corresponding column of $\tilde{\mathbf{U}}$ be quantized using exactly one bit-width. (3) The "bit-width selection constraint" guarantees that only permissible bit-widths are utilized for quantization. The variable $f \in \mathbb{R}^{1 \times N_b}$ denotes the set of admissible bit-widths, where $f_{0,k} = 1$ indicates that the $k^{\text{th}}$ bit-width in $Q$ is allowable. (4) The "bit-width number constraint" restricts the number of admissible bit-widths to a maximum of $f_{\max}$.

The 0/1 integer linear programming optimization problem is then solved with the CVXPY (Diamond & Boyd, 2016) library and the SCIP (Maher et al., 2016) solver. We report the optimization solving time in Appendix E.4, which costs 29.4 minutes for Qwen2.5-Math-7B-Instruct. This overhead is acceptable, as the model requires quantization only once. By solving Eq. (8), we obtain an optimal mixed-precision quantization scheme that minimizes the error while satisfying predefined bit budget constraints. This allows us to derive task-specific mixed-precision quantization strategies which balance the "scaling" and "difference" terms, leading to improved performance across various tasks.

## 4 EXPERIMENTS

### 4.1 EXPERIMENT SETUP

**Calibration Dataset.** Following Delta-CoMe (Ping et al., 2024), DELTAMIX randomly samples 128 examples, each containing 2048 tokens, from the C4 training set as the calibration dataset. This configuration is consistently applied across all calibration-dependent methods.

**Evaluation Tasks.** We evaluate our methods on four distinct tasks: reasoning, math, code generation, and multi-modal. These tasks encompass a vast array of current directions based on fine-tuning with open-source LLMs. **Reasoning:** We use the Math500 and AIME2024 datasets as the test set. **Math:** We use the GSM8K (Cobbe et al., 2021) and Math500 (Lightman et al., 2023) datasets as the test set. **Code Generation:** We use HumanEval (Chen et al., 2021) and MBPP (Austin et al., 2021) as the test set. **Multi-Modal:** We utilize the GQA (Hudson & Manning, 2019) and the image part of ScienceQA (Lu et al., 2022) datasets. Please refer to Appendix D.1 for more details.

**Models.** To ensure a comprehensive comparison, we evaluate both 7B and 13-14B models across the four tasks with various backbones. See Table 10 in Appendix D.1 for more details about the backbones and aligned models used. During inference, we employ a greedy search strategy.

**Baselines.** We compare DELTAMIX with three baselines: SVD-based low-rank compression (Ryu et al., 2023), BitDelta (Liu et al., 2024), Delta-CoMe (Ping et al., 2024) at compression ratio $1/\alpha = 16$. All methods are evaluated using NVIDIA L20 GPUs.

### 4.2 MAIN RESULTS

Tables 1 and 2 present the results of DELTAMIX on both the 7B and 13-14B models across four tasks, in comparison to the baselines. Notably, DELTAMIX demonstrates superior overall performance on

Table 1: Comparison of DELTAMIX and baselines on various tasks across 7B-sized models. We report the results in the format "mean(std)" with three runs for Delta-CoMe and DELTAMIX.

| Method | $\alpha$ | DeepSeek-R1-Distill-Qwen | | Qwen2.5-Math-Instruct | | Qwen2.5-Coder-Instruct | | Qwen2.5-VL-Instruct | | AVG |
|---|---|---|---|---|---|---|---|---|---|---|
| | | Math500 | AIME2024 | Math500 | GSM8K | Humaneval | Mbpp | GQA | SQA | |
| Backbone | 1 | 70.6 | 16.7 | 70.6 | 84.8 | 72.0 | 80.7 | - | - | - |
| Aligned | 1 | 86.0 | 40.0 | 80.2 | 94.8 | 87.2 | 82.8 | 60.5 | 76.7 | 76.0 |
| Low-Rank | 1/16 | 72.2 | 13.3 | 59.6 | 70.3 | 84.1 | **86.2** | 0.0 | 0.0 | 48.2 |
| BitDelta | 1/16 | 1.4 | 0.0 | 71.2 | 84.0 | 83.5 | 83.9 | 0.0 | 0.3 | 40.5 |
| Delta-CoMe | 1/16 | 82.4(1.11) | 30.0(3.30) | 74.8(0.35) | 94.5(0.00) | 85.0(0.96) | 82.7(0.17) | 49.4(1.65) | 76.5(0.26) | 71.9 |
| DELTAMIX | 1/16 | **82.7(0.83)** | **36.7(3.35)** | **77.7(1.03)** | **94.6(0.51)** | **85.6(0.35)** | 83.1(0.25) | **52.4(2.30)** | **79.4(0.83)** | **74.0** |

Table 2: Comparison of DELTAMIX and baselines on various tasks across 13-14B-sized models. We report the results in the format "mean(std)" with three runs for Delta-CoMe and DELTAMIX.

| Method | $\alpha$ | DeepSeek-R1-Distill-Qwen | | MetaMath | | Qwen2.5-Coder-Instruct | | LLAVA-V1.5 | | AVG |
|---|---|---|---|---|---|---|---|---|---|---|
| | | Math500 | AIME2024 | Math500 | GSM8K | Humaneval | Mbpp | GQA | SQA | |
| Backbone | 1 | 76.4 | 3.3 | 1.8 | 4.3 | 78.7 | 84.7 | - | - | - |
| Aligned | 1 | 87.4 | 40.0 | 22.6 | 71.0 | 90.2 | 85.4 | 63.3 | 72.8 | 66.6 |
| Low-Rank | 1/16 | 57.2 | 6.7 | 15.8 | 64.0 | 86.6 | **88.6** | 57.0 | 71.4 | 55.9 |
| BitDelta | 1/16 | 82.8 | 23.3 | 22.4 | 65.8 | 89.0 | 86.5 | 61.2 | **73.0** | 63.0 |
| Delta-CoMe | 1/16 | 76.5(3.38) | 24.5(6.93) | **22.9(0.12)** | 70.2(0.56) | 90.6(0.75) | 86.5(0.70) | **62.8(0.09)** | 72.3(0.20) | 63.3 |
| DELTAMIX | 1/16 | **80.2(2.09)** | **31.1(3.81)** | 21.7(0.64) | **71.2(0.26)** | **91.5(0.60)** | 86.9(0.12) | 62.7(0.04) | 72.1(0.18) | **64.7** |

both the 7B and 13-14B models, surpassing the best baseline, Delta-CoMe, by an average of 2.9% and 2.2%, respectively.

When analyzing the various tasks, we observe that DELTAMIX exhibits more pronounced improvements in challenging scenarios characterized by a significant performance gap between the baseline methods and the aligned model. This is particularly evident in reasoning-intensive benchmarks, such as AIME2024, as well as in multimodal tasks utilizing 7B backbones. For instance, DELTAMIX surpasses the previous state-of-the-art model, Delta-CoMe, by 22.3% on the 7B model and by 26.9% on the 14B model. Further analysis reveals that these models display larger norms for $\Delta\mathbf{W}$. Specifically, the median norm of DeepSeek-R1-Distill-Qwen-7B and Qwen2.5-VL-Instruct is 6.5 and 10.3 times that of Qwen-Coder-Instruct-7B, with corresponding values of 26.13 and 41.45 compared to 4.02, respectively. In this context, baseline methods struggle to achieve optimal solutions due to their empirical nature. In contrast, DELTAMIX directly optimizes quantization error from a mathematical perspective, enabling it to fully leverage its strengths in demanding tasks. However, on tasks where baselines already achieve near-lossless accuracy, such as MBPP and HumanEval on the 7B backbone, DELTAMIX performs comparably to the best baseline. In these scenarios, the norm of $\Delta\mathbf{W}$ is relatively small and can be easily compressed, leading to a ceiling effect: $\Delta\mathbf{W}$ can be quantized almost losslessly by existing baselines, leaving little room for further improvement.

We also compare the quantization time cost of DELTAMIX and Delta-CoMe. Please refer to Appendix E.4 for more details. The results show that DELTAMIX (resp. Delta-CoMe) requires only 1.2 (resp. 0.4)hours for 7B models and 2.4 (resp. 0.8) hours for 14B models on a single GPU. Although DELTAMIX is slower than Delta-CoMe, the time cost remains acceptable since the quantification process needs to be performed only once.

## 4.3 COMPARE WITH BROADER BASELINES

To further validate the effectiveness of DELTAMIX, we introduced two additional baselines: SVD-LLM (Wang et al., 2025) and the sparse-quant method, SpQR (Dettmers et al., 2023), to compare with DELTAMIX on the 7B-sized models. Considering that SpQR quantizes Zeros and Scales to save space for storing some outliers in 32 bits, we divide SpQR into two baselines: 1) No quantization of Zeros and Scales, but no outliers stored. 2) Using a two-step quantization method, storing some

Table 3: Comparison of DELTAMIX and boarder baselines on various tasks across 7B-sized models. As the SpQR method integrates sparsity and quantization, we divide this method into two baselines, one with and one without outliers.

| Method | $\alpha$ | DeepSeek-R1-Distill-Qwen | | Qwen2.5-Math-Instruct | | Qwen2.5-Coder-Instruct | | Qwen2.5-VL-Instruct | | AVG |
|---|---|---|---|---|---|---|---|---|---|---|
| | | MATH500 | AIME2024 | Math500 | GSM8K | Humaneval | Mbpp | GQA | SQA | |
| SVD-LLM | 1/16 | 32.8 | 10.0 | 67.4 | 82.8 | 85.2 | **83.1** | 0.0 | 0.0 | 45.2 |
| SpQR(No Outliers) | 1/16 | 2.4 | 0.0 | 12.6 | 38.5 | 84.8 | 78.3 | 0.0 | 0.0 | 27.0 |
| SpQR(0.01% Outliers) | 1/16 | 45.0 | 10.0 | 71.2 | 89.2 | 85.4 | 82.3 | 0.0 | 0.0 | 48.0 |
| DELTAMIX | 1/16 | **82.7** | **36.7** | **77.7** | **94.6** | **85.6** | 83.1 | **52.4** | **79.4** | **74.0** |

Table 4: The performance of DELTAMIX to quantize Qwen2.5-Math-7B-Instruct with different number of calibration data.

| Calibration Size | Math500 | GSM8K | Average |
|---|---|---|---|
| 16 | 76.4 | 94.5 | 85.5 |
| 32 | 76.2 | **95.1** | 85.7 |
| 64 | 76.8 | 94.3 | 85.6 |
| 128 | **77.6** | 94.8 | **86.2** |
| 256 | 76.0 | 94.1 | 85.1 |

Table 5: The performance of DELTAMIX to quantize Qwen2.5-Math-7B-Instruct using calibration data drawn from C4 and Wikitext2.

| | Math500 | GSM8K | Average |
|---|---|---|---|
| C4 | **77.6** | **94.8** | **86.2** |
| Wikitext2 | 76.6 | **94.8** | 85.7 |
| MetaMath | 75.4 | 93.6 | 84.5 |

outliers. The results in Table 3 indicate that DELTAMIX consistently outperforms all three baselines. In particular, for Qwen2.5-VL-Instruct, except for DELTAMIX, all baselines have lost its multimodal capability.

## 4.4 ABLATION OF CALIBRATION DATASET

Since DELTAMIX is a calibration-dependent method, to verify its robustness on calibration, we conducted experiments with different sizes and domains of the calibration dataset to quantize Qwen2.5-Math-7B-Instruct. For calibration on domains, each calibration set contains 128 randomly sampled sequences of length 2048. Due to the insufficient number of sequences of this length in the Meta-MathQA dataset, we concatenated multiple question–answer pairs in a few-shot format. To examine the effect of dataset size on calibration, we varied the number of calibration samples from 16 to 256. The results in Tables 4 and 5 demonstrate that DELTAMIX performs well on all calibration setups, confirming DELTAMIX 's robustness.

## 4.5 ANALYSIS OF $f_{\max}$

In DELTAMIX, we set a hyperparameter termed $f_{\max}$ to constrain the number of active bitwidths during quantization. This section examines the performance of DELTAMIX under varying values of $f_{\max}$. As shown in Table 6, DELTAMIX consistently achieves better performance than Delta-CoMe across all settings, indicating that DELTAMIX is insensitive to the choice of $f_{\max}$. In the main experiment, we set $f_{\max}$ to 4 to be consistent with Delta-CoMe.

Table 6: Performance across different $f_{\max}$. We report the results in the format "mean(std)" with three runs.

| Method | $f_{\max}$ | DeepSeek-R1-Distill-Qwen-14B | | AVG |
|---|---|---|---|---|
| | | Math500 | AIME2024 | |
| Delta-CoMe | - | 76.5(3.38) | 24.5(6.93) | 50.5 |
| DELTAMIX | 2 | **80.7(1.75)** | **33.3(3.35)** | **57.0** |
| | 3 | 79.9(1.53) | 30.0(8.83) | 55.0 |
| | 4 | 80.2(2.09) | 31.1(3.81) | 55.7 |
| | 5 | 79.5(0.99) | **33.3(6.65)** | 56.4 |
| | 6 | 79.5(2.21) | **33.3(3.35)** | 56.4 |

## 4.6 ABLATION OF RTC

We conducted experiments to assess the necessity of RTC, as detailed in Table 7. Overall, RTC consistently enhances our method, yielding an average performance improvement of 2.2%. The results indicate that mitigating the deviation in the quantization loss of $\mathbf{U}$ enables DELTAMIX to retain more information from $\Delta\mathbf{W}$.

Table 7: Performance ablation of RTC. We report the results in the format "mean(std)" with three runs.

| | LLAVA-V1.5 | | DeepSeek-R1-Distill-Qwen-14B | | AVG |
|---|---|---|---|---|---|
| | GQA | SQA | Math500 | AIME2024 | |
| Delta-CoMe | 62.8(0.09) | **72.3(0.20)** | 76.5(3.38) | 24.5(6.93) | 59.0 |
| DELTAMIX | 62.7(0.04) | 72.1(0.18) | **80.2(2.09)** | **31.1(3.81)** | 61.5 |
| DELTAMIX (W/O RTC) | **62.8(0.02)** | 72.2(0.05) | 78.2(0.28) | 27.5(3.81) | 60.2 |

The importance of RTC is particularly pronounced in challenging tasks; for instance, it improves performance by 13.1% on the AIME2024 task. This improvement can be attributed to the more substantial quantization errors associated with quantizing $\mathbf{V}$ in these cases, thereby highlighting the critical need for reconstruction target correction.

In Appendix E.2, we evaluate the required time of RTC across four model sizes to demonstrate the high efficiency of the RTC. The results show that, for a 14B model, RTC requires only 1.35s to process a transformer block, accounting for merely 1.18% of the total quantization time. Please refer to Appendix E.2 for more details.

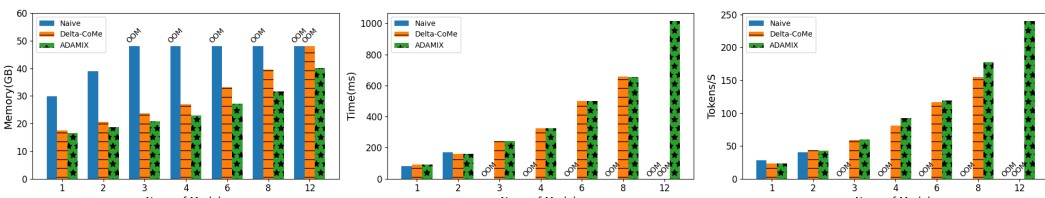

Figure 3: End-to-end decoding latency evaluation with varying numbers of deployed models using Qwen2.5-7B variants. (Left) Decoding memory usage. (Middle) Prefill time. (Right) Generation speed.

## 5 ANALYSES

### 5.1 INFERENCE SPEED AND MEMORY COST

Following the setup of Liu et al. (2024), we evaluate the end-to-end decoding latency of Qwen2.5-7B variants using a single L20 GPU. As shown in Figure 3, we consider the setting where each deployed model receives one distinct request simultaneously—e.g., 12 deployed models correspond to a batch size of 12- with latency evaluation in three perspectives: (1) Memory Usage: This one measures peak GPU memory usage during concurrent inference, accounting for both model parameters and activation storage. (2) Prefill Time: This part focuses on the time the models take to process user-input prompts. Each request contains 512 input tokens, and we report the time (in ms) the model takes to handle them. (3) Generation Speed: This part evaluates how quickly the model generates output tokens (tokens/s) for each request. Since the prefill time already measures prompt processing, each request starts from the "[BOS]" token and generates 512 tokens sequentially.

As shown in Figure 3 (left), a single GPU can deploy only two aligned models simultaneously. In contrast, it can support up to 8 and 12 models concurrently for Delta-CoMe and DELTAMIX, respectively. This enhancement is attributable to the fact that, as the number of models increases, both methods necessitate only the additional deployment of compressed delta weights, thereby significantly reducing memory overhead. Notably, while Delta-CoMe exhausts GPU memory at 12 models, DELTAMIX does not. Our further analysis indicates that DELTAMIX typically employs fewer ranks, namely allocates a greater number of singular vectors with a bid-width of 0, thereby enhancing the GPU memory utilization efficiency.

For the end-to-end decoding latency illustrated in Figure 3 (middle, right), we find that Delta-CoMe and DELTAMIX introduce overhead to Naive when the number of deployed model is small. However, Delta-CoMe and DELTAMIX scale better and effectively translate the saved GPU memory into improved decoding latency. In contrast, the Naive approach quickly encounters out-of-memory issues. Furthermore, DELTAMIX exhibits a superior generation speed compared to Delta-CoMe at scale, while the prefill times for both methods remain comparable. In Appendix E.2, we conduct more latency evaluation under varying arrival rates and request distributions following (Yao et al., 2024).

### 5.2 DELTA-COMPRESSION VS. DELTA-TUNING

Delta-compression decomposes the delta weights of a fully fine-tuned model into low-rank and low-bit representations, thereby reducing storage and inference costs. Delta-tuning methods, such as LoRA, are closely related to delta-compression but primarily aim to reduce the training costs of LLMs while achieving performance comparable to that of full fine-tuning. However, in various tasks—particularly more complex ones like code and math tasks—delta-tuning methods tend to underperform full fine-tuning (Biderman et al., 2024). This suggests that relying solely on delta-tuning may be insufficient.

In this section, we train the DeepSeek-LLM-7B-Base (DeepSeek-AI, 2024) on math and code tasks using both LoRA and full fine-tuning. We subsequently apply DELTAMIX to the delta weights of the fully fine-tuned model and LoRA.

Table 8: Performance comparison between Delta-Compression and LoRA. Aligned is full fine-tuned model. For DELTAMIX, we report the results in the format "mean(std)" with three runs.

| Method | $\alpha$ | Code | | Math | | AVG |
|---|---|---|---|---|---|---|
| | | Humaneval | Mbpp | Math500 | GSM8K | |
| Backbone | 1 | 24.4 | 46.0 | 3.8 | 14.7 | 22.2 |
| Aligned | 1 | 46.3 | 48.9 | 14.6 | 58.3 | 42.0 |
| LoRA | 1/16 | 34.1 | 47.7 | 9.4 | 50.9 | 35.5 |
| DELTAMIX | 1/16 | **43.3(0.6)** | **50.2(0.82)** | **13.5(0.76)** | **56.1(0.82)** | **40.8** |
| DELTAMIX-LoRA | 1/64 | 34.1 | 47.6 | 10.4 | 49.6 | 35.4 |
| DELTAMIX | 1/64 | **39.0** | **51.6** | **11.6** | **54.3** | **39.1** |

Additional experimental details can be found in Appendix D.2. Table 8 presents a comparison of DELTAMIX with LoRA. The results indicate that DELTAMIX consistently outperforms LoRA across all tasks. DELTAMIX achieves an average score of 40.8, which is close to the aligned model's score of 42.0, representing a 14.9% improvement over LoRA.

Furthermore, applying DELTAMIX to LoRA can further improve the compression ratio without sacrificing performance. Table 8 shows that the average performance difference of LoRA before and after compression is 0.01. Notably, Baselines like BitDelta and Delta-CoMe cannot apply to LoRA. BitDelta directly quantizes $\Delta\mathbf{W}$ to 1 bit without employing any low-rank approximation. Consequently, it cannot effectively utilize the low-rank properties inherent in LoRA. For Delta-CoMe, the empirically determined mixed-precision scheme is fixed and does not offer a clear method for allocating mixed precision at other compression ratios. In contrast, DELTAMIX allows compression of $\Delta\mathbf{W}$ to arbitrary ratios, making it more flexible and practically advantageous.

## 5.3 ANALYZING QUANTIZATION ERROR

To better understand the difference between various delta-compression methods, we compute the quantization error on Qwen2.5-Math-7B-Instruct model as defined in Equation (1). Since outliers play a critical role in model compression (Dettmers et al., 2023; Lin et al., 2024), we also report the average error for the top 1% of activations with the largest absolute values in the aligned model, categorizing them as outliers. As different layers contribute differently to the final output (Wu et al., 2024), we categorize the first 9 layers, layers 9 to 17, and the last 10 layers as low, mid, and high groups, respectively, and report the average error of each group. See Table 20 of Appendix E.8 for more details.

Table 9: Average quantization error ($\times$ 1e2) on Qwen2.5-Math-7B-Instruct model with Eq. (1)."Low", "Mid", and "High" denote the first 9 layers, layers 9 to 17, and the last 10 layers, respectively. "All" and "Out" denote the average error across all activations and the average error of the top 1% of activations.

|  | Low | | Mid | | High | |
|---|---|---|---|---|---|---|
|  | All | Out | All | Out | All | Out |
| Low-Rank | 1.82 | 3.67 | 1.50 | 2.84 | 21.12 | 1890.34 |
| BitDelta | 2.18 | 2.81 | **0.61** | **1.08** | 21.51 | 3162.58 |
| Delta-CoMe | 0.76 | 1.79 | 0.75 | 1.33 | 7.54 | 470.82 |
| DELTAMIX | **0.66** | **1.46** | 0.66 | 1.12 | **6.81** | **426.20** |

As demonstrated in Table 9, DELTAMIX consistently exhibits lower overall quantization error compared to all baseline methods, attributable to its inherent objective of minimizing quantization error. In the mid layers, DELTAMIX shows a slightly higher error than BitDelta, with values of 0.66 versus 0.61 for all activations and 1.12 versus 1.08 for outlier activations, respectively. However, it is important to note that since BitDelta is an empirical method, it cannot guarantee low quantization error across all layers. For example, in the high layers, BitDelta exhibits significantly higher error rates compared to DELTAMIX, with values of 21.51 versus 6.81 for all activations and 3162.58 versus 426.20 for outlier activations, respectively. These experiments further illustrate that DELTAMIX effectively reduces quantization error, thereby preserving the information contained in the delta weights as much as possible. In Appendix E.7, we visualize the bit allocation results of DELTAMIX across different weight types and layers using the Qwen2.5-Math-7B-Instruct model.

## 6 CONCLUSION

In this study, we present DELTAMIX, an adaptive mixed-precision delta-compression framework aimed at minimizing quantization error in the SVD space without introducing additional assumptions. DELTAMIX offers a theoretical proof of the necessity for mixed-precision delta-compression and provides a practical quantization solution that involves solving a 0/1 linear integer programming problem and employing a reconstruction target correction method. DELTAMIX outperforms all baseline delta-compression methods across four distinct downstream tasks, including reasoning, math, code, and multi-modal tasks, utilizing eight widely adopted aligned LLMs with backbone pre-trained models, including Qwen2.5, Qwen2.5-Math, Qwen2.5-Coder, and LLaMA2. Moreover, DELTAMIX significantly reduces deployment costs by minimizing memory overhead and accelerating inference. We believe that DELTAMIX provides considerable theoretical and practical value, particularly in scenarios involving multi-tenant deployments.

ETHICS STATEMENT

We propose an adaptive mixed-precision delta-compression framework designed to minimize quantization error in the singular value decomposition space. Our experiments rely exclusively on publicly available datasets and models, without involving human subjects or sensitive data. We do not anticipate any direct negative consequences arising from this approach.

REPRODUCIBILITY STATEMENT

To facilitate reproducibility, we describe our experimental setup in Section 4.1 and provide additional details, including models, datasets, metrics, and GPUs, in Appendix D. Furthermore, our implementation is publicly available at https://anonymous.4open.science/r/ICLR-Annoymous-CD59.

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

## A   Limitation and Broader Impact

DELTAMIX significantly reduces hardware requirements and computational costs for serving multiple finetuned models, thereby enabling smaller entities to deploy advanced large language models more feasibly. Additionally, it lowers power consumption and reduces the carbon emissions associated with LLM deployment. Despite DELTAMIX 's demonstrated improvements over baseline methods in reducing the performance gap between compressed and aligned models, it is important to note that DELTAMIX remains a lossy compression method for certain tasks. We believe this is an important consequence and encourage future research to further minimize this performance gap, particularly in tasks where performance degradation is substantial.

## B   LLMs Usage

In this work, large language models (LLMs) were used solely as auxiliary tools for grammar correction and text refinement.

## C   Formula Derivation

### C.1   V Hessian Matrix

$$
\begin{aligned}
&d_{\hat{\mathbf{V}}}^2 \left\| \mathbf{U\Sigma V}X - \mathbf{U\Sigma}\hat{\mathbf{V}}X \right\|_F^2 \\
&= 2tr(\mathbf{U\Sigma d\hat{V}}XX^T\mathbf{d\hat{V}^T\Sigma^T U^T}) \\
&= 2tr(\mathbf{\Sigma^T U^T U\Sigma d\hat{V}}XX^T\mathbf{d\hat{V}^T}) \\
&= 2(d\,\mathrm{vec}(\hat{V}^T))^T(\mathbf{\Sigma^T\Sigma} \otimes XX^T)(d\,\mathrm{vec}(\hat{V}^T)) \\
&= 2(d\,\mathrm{vec}\hat{V})^T(\mathbf{\Sigma^T\Sigma} \otimes XX^T)(d\,\mathrm{vec}(\hat{V})) \\
&\Rightarrow \mathbf{H^V} = 2\mathbf{\Sigma^T\Sigma} \otimes XX^T \\
&\Rightarrow \mathbf{H}_i^{\mathbf{V}} = 2\Sigma_{ii}^2 \cdot XX^T
\end{aligned}
\tag{9}
$$

Here $\otimes$ denotes the Kronecker product.

### C.2   U Hessian Matrix

$$
\begin{aligned}
&d_{\hat{\mathbf{U}}}^2 \left\| \mathbf{U\Sigma}\hat{\mathbf{V}}X - \hat{\mathbf{U}}\mathbf{\Sigma}\hat{\mathbf{V}}X \right\|_F^2 \\
&= d\hat{\mathbf{U}}\mathbf{\Sigma}\hat{\mathbf{V}}XX^T\hat{\mathbf{V}}^T\mathbf{\Sigma^T}d\hat{\mathbf{U}}^T \\
&= X^T\hat{\mathbf{V}}^T\mathbf{\Sigma^T}d\hat{\mathbf{U}}^Td\hat{\mathbf{U}}\mathbf{\Sigma}\hat{\mathbf{V}}X \\
&= (d\,\mathrm{vec}\hat{U})^T\mathbf{K_{rh_{out}}}(\mathbf{I} \otimes \mathbf{\Sigma}\hat{\mathbf{V}}XX^T\hat{\mathbf{V}}^T\mathbf{\Sigma^T})\mathbf{K_{h_{out}r}}(d\,\mathrm{vec}\hat{U}) \\
&= 2(d\,\mathrm{vec}\hat{U})^T(\mathbf{I} \otimes \mathbf{\Sigma}\hat{\mathbf{V}}XX^T\hat{\mathbf{V}}^T\mathbf{\Sigma^T})(d\,\mathrm{vec}\hat{U}) \\
&\Rightarrow \mathbf{H}_i^{\mathbf{U}} = \mathbf{H^U} = 2\mathbf{\Sigma}\hat{\mathbf{V}}XX^T\hat{\mathbf{V}}^T\mathbf{\Sigma^T}
\end{aligned}
\tag{10}
$$

Here $\mathbf{K_{h_{out}r}}$ is the commutation matrix, and $\mathbf{K_{h_{out}r}^{-1}} = \mathbf{K_{rh_{out}}}$.

### C.3   Detailed Derivation Process for new U

$$
\begin{aligned}
&d_{\tilde{\mathbf{U}}} \left\| \mathbf{U\Sigma V}X - \tilde{\mathbf{U}}\mathbf{\Sigma}\hat{\mathbf{V}}X \right\|_F^2 \\
&= 2tr(d\tilde{\mathbf{U}}\mathbf{\Sigma}\hat{\mathbf{V}}X(\tilde{\mathbf{U}}\mathbf{\Sigma}\hat{\mathbf{V}}X - \mathbf{U\Sigma V}X)^T) \\
&= 2tr(\mathbf{\Sigma}\hat{\mathbf{V}}X(\tilde{\mathbf{U}}\mathbf{\Sigma}\hat{\mathbf{V}}X - \mathbf{U\Sigma V}X)^Td\tilde{\mathbf{U}}) \\
&\Rightarrow \frac{\partial\mathbb{L}}{\partial\tilde{\mathbf{U}}} = (\tilde{\mathbf{U}}\mathbf{\Sigma}\hat{\mathbf{V}}X - \mathbf{U\Sigma V}X)X^T\hat{\mathbf{V}}^T\mathbf{\Sigma^T}
\end{aligned}
\tag{11}
$$

Table 10: Selected backbone and aligned models for the examined four tasks.

| Task | 7B Models | | 13-14B Models | |
|------|-----------|--|---------------|--|
| | Backbone | Aligned | Backbone | Aligned |
| Math | Qwen2.5-Math | Qwen2.5-Math-Instruct | LLaMA2 | MetaMath |
| Reasoning | Qwen2.5-Math | DeepSeek-R1-Distill-Qwen | Qwen2.5 | DeepSeek-R1-Distill-Qwen |
| Coder | Qwen2.5-Coder | Qwen2.5-Coder-Instruct | Qwen2.5-Coder | Qwen2.5-Coder-Instruct |
| Multi-Modal | Qwen2.5 | Qwen2.5-VL-Instruct | LLaMA2 | LLAVA-V1.5 |

By setting the gradient of the loss to zero, DELTAMIX gets the corrected $\tilde{\mathbf{U}}$ as follow:

$$
\frac{\partial \mathbb{L}}{\partial \tilde{\mathbf{U}}} = (\tilde{\mathbf{U}}\mathbf{\Sigma}\hat{\mathbf{V}}X - \mathbf{U}\mathbf{\Sigma}\mathbf{V}X)X^T\hat{\mathbf{V}}^{\mathbf{T}}\mathbf{\Sigma}^{\mathbf{T}} = 0
$$
$$
\Rightarrow \tilde{\mathbf{U}} = \mathbf{U}\mathbf{\Sigma}\mathbf{V}XX^T\hat{\mathbf{V}}^{\mathbf{T}}\mathbf{\Sigma}^{\mathbf{T}}(\mathbf{\Sigma}\hat{\mathbf{V}}XX^T\hat{\mathbf{V}}^{\mathbf{T}}\mathbf{\Sigma}^{\mathbf{T}})^{-1}
$$

(12)

# D EXPERIMENTS SETUP

## D.1 MAIN EXPERIMENTS

We evaluate our methods across models in Table 10 on four distinct tasks: math, reasoning, code generation, and multi-modal. These tasks encompass a vast array of current directions based on fine-tuning with open-source LLMs.

• **Math.** We use the GSM8K (Cobbe et al., 2021) and Math500 (Lightman et al., 2023) datasets as the test set. We follow the prompt format of WizardMath (Luo et al., 2025) and set the maximum generation length to 1024. The evaluation metric is accuracy, determined by comparing the model-generated solution to the ground truth.

• **Reasoning.** We use the Math500 and AIME2024 datasets as the test set. For the reasoning prompt of AIME2024, we follow with (Naman Jain & et al., 2024). The maximum length of both tasks is set to 8192. The evaluation metric is accuracy, determined by comparing the model-generated solution to the ground truth.

• **Code Generation.** We use two widely used datasets as the test set: HumanEval (Chen et al., 2021) and MBPP (Austin et al., 2021). We follow the Magicoder (Wei et al., 2024) evaluation framework for HumanEval and adopt EvalPlus (Liu et al., 2023a) for MBPP. The evaluation metric is the pass rate (pass@1), which measures whether the code generated in a single attempt successfully passes the test cases.

• **Multi-Modal.** We utilize the GQA (Hudson & Manning, 2019) and the image part of ScienceQA (Lu et al., 2022) datasets, both commonly used for evaluating VLM performance, as our test set. We adopt lmms-eval (Zhang et al., 2024) to evaluate both tasks. The evaluation metric is accuracy, which measures whether the model selects the correct option.

To accelerate DELTAMIX's quantization, we discard the last $k$ ranks of $\mathbf{V}$, where $k = \left\lfloor \frac{G_b(h_{\text{in}} \cdot h_{\text{out}})}{(h_{\text{in}} + h_{\text{out}}) \cdot \text{bit}_{\min}} \right\rfloor$. Here, $\text{bit}_{\min}$ denotes the smallest non-zero bit-width allowed in quantization. Our acceleration scheme, which eliminates the singular components with the smallest singular values, sacrifices some performance in exchange for reduced computational costs. To validate this approach, we conducted experiments on Qwen2.5-Math-7B-Instruct by discarding between

Table 11: The performance of applying DELTAMIX to Qwen2.5-Math-7B-Instruct with discard the last $k$% ranks.

| Drop Ratio | Math500 | GSM8K | Average | Costing Time |
|------------|---------|-------|---------|--------------|
| 0 | 77.6 | 94.8 | 86.2 | 1.29h |
| 0.1 | 75.4 | 94.2 | 84.8 | 1.13h |
| 0.2 | 78.6 | 94.4 | 86.5 | 1.13h |
| 0.3 | 75.6 | 94.2 | 84.9 | 1.00h |
| 0.4 | 75.8 | 94.3 | 85.1 | 1.00h |
| 0.5 | 76.0 | 93.7 | 84.9 | 1.00h |
| 0.6 | 74.8 | 94.0 | 84.4 | 0.94h |

0% and 60% of the trailing singular components. As shown in Table 11, DELTAMIX performs optimally at low drop ratios (0% and 20%), confirming that this technique is primarily aimed at enhancing speed. As we discard more trailing singular components, performance declines; however, this slight decrease can be exchanged for an improvement in speed.

## D.2 DELTA-COMPRESSION VS. DELTA-TUNING

Specifically, we set the LoRA rank to 128 and the scale factor to 128, training LoRA for all model parameters for 3 epochs using a cosine schedule with a peak learning rate of 4e-5 and a warm-up ratio

of 0.1, using model deepseek-llm-7b-base (DeepSeek-AI, 2024). We randomly sample 50k training examples from MetaMathQA (Yu et al., 2023) and Magicoder-Evol-Instruct (Wei et al., 2024) for the math and code tasks, respectively. To ensure a fair comparison, we fine-tune all model parameters using the same datasets as those used for LoRA training. We then apply DELTAMIX to both math and code finetuned LLMs.

# E   MORE EXPERIMENTS

## E.1   ANALYZING THE DIFFERENT QUANTIZATION SCHEMES IN **U**

In this section, we investigate the effect of applying different quantization schemes to **U** in order to assess the necessity of mixed precision. Our evaluation is conducted on Qwen2.5-Math-7B-Instruct. The results show that there is no significant difference between DELTAMIX and other quantization methods for **U**. As shown in Table 12, "x-bit" denotes quantization of **U** with x-bit precision. The "DELTAMIX-row" setting applies the optimization model to determine the scheme and performs quantization in

Table 12: We evaluate the performance of various quantization schemes applied to **U** on Qwen2.5-Math-7B-Instruct. Here, "x-bit" denotes quantization of **U** at x-bit precision. The "DELTAMIX-row" setting refers to applying the optimization model to determine the scheme and performing quantization in a row-wise manner, whereas "DELTAMIX " indicates employing the same quantization scheme used for **V**, with quantization carried out column by column.

|  | $\alpha$ | Math500 | GSM8K | AVG |
|---|---|---|---|---|
| **U**(2bit),**V**(DELTAMIX) | 1/16 | 76.8 | 93.6 | 85.2 |
| **U**(3bit),**V**(DELTAMIX) | 1/16 | 75.6 | 93.4 | 84.5 |
| **U**(DELTAMIX-row),**V**(DELTAMIX) | 1/16 | 75.2 | 93.6 | 84.4 |
| DELTAMIX | 1/16 | 75.2 | 93.9 | 84.6 |

a row-wise manner, whereas "DELTAMIX" adopts the same quantization scheme as **V** and conducts quantization column by column. The performance differences across schemes are minimal, with the largest gap in average scores being only 0.95%, observed between the "DELTAMIX-row" setting and the 2-bit quantization. These results suggest that the choice of quantization strategy for **U** has only a limited impact on overall performance.

## E.2   INFERENCE SPEED AND MEMORY COST

To demonstrate the impact of DELTAMIX on inference speed and memory cost, we implement a simple Triton  (Tillet et al., 2019) kernel for DELTAMIX. We compare our kernel with naive aligned models. Since there is no packing function of Delta-CoMe, we use our packing function and kernel for the Delta-CoMe method.

Following the setup in Yao et al. (2024), we assess the end-to-end system performance under varying arrival rates and request distributions. We consider two types of model popularity distribution: 1) Uniform: all models are equally popular. 2) Skewed: model popularity follows a Zipf-$\alpha$ distribution.

We evaluate the performance when serving 32 model variants of Qwen2.5-7B. Requests are sent to the serving system at a variable Poisson arrival rate ($\lambda$). To simplify, each request consists of 512 tokens, with the model generating one token as its response. We run the simulations for 100 seconds across different arrival rates and model distributions, measuring performance using two metrics: 1) end-to-end latency averaged over all requests; 2) Throughput, number of requests processed per second. All experiments are conducted on a single L40 GPU, with 28G of memory for storing models and the remaining memory for inference.

Table 13: The Throughput and End-to-end system performance under varying arrival rates and request distributions when serving 32 model variants of Qwen2.5-7B.

|  | $\lambda = 0.5$ | | $\lambda = 1.0$ | |
|---|---|---|---|---|
|  | Throughput(req/s) | E2E(s) | Throughput(req/s) | E2E(s) |
| Zipf ($\alpha = 1.5$) | | | | |
| Naive | 0.21 | 52.42 | 0.18 | 198.48 |
| Delta-CoMe | **0.42** | 0.55 | **0.87** | 0.68 |
| DELTAMIX | **0.42** | **0.52** | **0.87** | **0.62** |
| Uniform | | | | |
| Naive | 0.07 | 253.93 | 0.08 | 481.42 |
| Delta-CoMe | **0.42** | 0.81 | **0.86** | 1.44 |
| DELTAMIX | **0.42** | **0.79** | **0.86** | **1.17** |

As shown in the Table 13, DELTAMIX improves the throughput 6x and decreases end-to-end 100x compared to the naive method, because rather than loading the whole full-precision parameters, DELTAMIX quantizes the delta-parameters so that a GPU can load more delta-parameters and switch them easily between CPU and GPU.

### E.3 TIME REQUIRED FOR RTC

In this section, we evaluate the efficiency of the RTC method by measuring its time for processing a transformer block across four models of different sizes and comparing it to the corresponding proportion of the total quantization time. As shown in the Table 14, RTC processes an entire transformer block for a 70B model in just 8.16s, accounting for only 2.82% of the total quantization time. Referring to Table 7, RTC improves performance on the AIME2024 task by 13.1%. This demonstrates that RTC can improve the performance of more difficult-to-quantize models in a short time.

Table 14: The time (in seconds) consumed by applying RTC and quantizing a transformer block to four different-sized models.

| Model | RTC | Total Time | RTC (in percentage) |
|---|---|---|---|
| DeepSeek-R1-Distill-Qwen-1.5B | 0.15 | 29.87 | 0.50% |
| DeepSeek-R1-Distill-Qwen-7B | 1.35 | 99.25 | 1.36% |
| DeepSeek-R1-Distill-Qwen-14B | 1.35 | 114.88 | 1.18% |
| DeepSeek-R1-Distill-Llama-70B | 8.16 | 289.69 | 2.82% |

### E.4 TIME FOR QUANTIZATION

Table 15: Time cost (in seconds) for "Simulation", "Optimization", and "Quantization" for one transformer block on the Qwen2.5-Math-7B-Instruct model, which consists of 28 blocks.

| | | Simulation | Optimization | Quantization | Total |
|---|---|---|---|---|---|
| Delta-CoMe | Q_proj | 0.0 | | 3.6 | 50.5 |
| | K_proj | 0.0 | 0.0 | 3.6 | |
| | V_proj | 0.0 | | 3.6 | |
| | O_proj | 0.0 | 0.0 | 5.1 | |
| | Up_proj | 0.0 | 0.0 | 4.5 | |
| | Gate_proj | 0.0 | | 4.5 | |
| | Down_proj | 0.0 | 0.0 | 25.6 | |
| DELTAMIX | Q_proj | 4.7 | | 0.5 | 143.6 |
| | K_proj | 4.7 | 8.5 | 0.5 | |
| | V_proj | 4.7 | | 0.5 | |
| | O_proj | 6.1 | 11.5 | 0.5 | |
| | Up_proj | 5.8 | 20.5 | 2.8 | |
| | Gate_proj | 5.8 | | 2.8 | |
| | Down_proj | 30.2 | 22.5 | 11.0 | |

In this section, we evaluate the quantization time of DELTAMIX and Delta-CoMe within a single transformer block. The fundamental distinction between the two methods lies in their mixed-precision quantization strategies for each linear layer. DELTAMIX determines the strategy by minimizing quantization loss, formulated as a 0/1 integer linear programming problem, but Delta-CoMe adopts an empirical and fixed strategy for all linear layers.

To clarify the computational overhead, we decompose the quantization time into three components. The first is "simulation time", which reflects the cost of estimating quantization loss under different bit-widths. The second is "optimization time", incurred when solving the 0/1 integer linear programming problem. The third is the "quantization time" itself, representing the cost of quantizing each linear layer according to the selected strategy.

**Required Time for Each Part.** In Table 15, we report the detailed results of different types of linear layer for one transformer block in Qwen2.5-Math-7B-Instruct, which contains 28 blocks in total. For Delta-CoMe, both simulation and optimization times are zero because its mixed-precision quantization strategy is predetermined and applied uniformly across all linear layers; consequently, the entire forward pass is accounted for within the quantization time. In contrast, DELTAMIX incurs additional simulation and optimization costs, which are higher for Up_proj, Gate_proj, and Down_proj due to their larger row or column dimensions. Specifically, simulation time increases with the number of columns, while optimization time grows with the number of rows. Notably,

DELTAMIX 's quantization time is shorter than that of Delta-CoMe, since the forward pass is already included in its simulation stage.

Overall, although DELTAMIX takes 3x more time than Delta-CoMe, it only requires 1.2 hours for 7B models and 2.4 hours for 14B models on a single L20 GPU, which is acceptable. In contrast to Delta-CoMe's degraded performance on the large norm of $\Delta \mathbf{W}$, DELTAMIX consistently achieves comparable or better results across all scenarios.

**Required Time under Various Numbers of Candidate Bit-widths.** By quantizing Qwen2.5-Math-7B-Instruct with different numbers of candidate bit-widths, we further analyze the time cost for each part of the quantization. The results in Table 16 demonstrate that the "Quantization" remains nearly constant. For Quantization, the parameters of any rank, the assigned bit width is fixed once the mixed-precision scheme is determined; thus, varying the range of candidate bit widths has a negligible effect. In contrast, the "Optimization" and "Simulation" are directly affected by the number of available bit widths. The bit-width range determines the simulation rounds and the size of the ILP solution space.

Table 16: The required time for each part across various numbers of candidate bit-widths to quantize Qwen2.5-Math-7B-Instruct.

| #Candidate Bit-widths | Simulation | Optimization | Quantization | Total Time |
|---|---|---|---|---|
| 3 | 22.29 | 30.80 | 18.39 | 71.48 |
| 4 | 26.05 | 35.01 | 21.42 | 82.48 |
| 5 | 28.41 | 39.68 | 19.78 | 87.87 |
| 6 | 32.18 | 46.40 | 19.88 | 98.46 |
| 7 | 37.46 | 55.14 | 20.71 | 113.31 |
| 8 | 39.66 | 63.00 | 20.71 | 123.37 |

**Optimization Time Scale with Layer Size.** We measured the total time and memory needed to solve the ILP for a single transformer block across four different-sized models. We also report the GPU memory consumption (in GB) and the hidden and intermediate sizes corresponding to the size of the model. The results in Table 17 demonstrate that ILP runtime is primarily determined by the hidden dimension of an individual linear layer. It should also be noted that, to satisfy open-source requirements, our experiments employed a slower open-source solver (SCIP). In practice, the use of faster commercial ILP solvers or a reduced set of candidate bit-widths can substantially accelerate ILP solving. Thus, the reported optimization times should be interpreted as a lower bound for real-world deployments. Regarding memory usage, DELTAMIX quantizes a 70B model within a single L20 GPU with 47.53GB, indicating that DELTAMIX is not resource-intensive and is therefore suitable for large-scale applications in resource-constrained environments or for parallel quantization of multiple models.

Table 17: The "Optimization" time, GPU memory usage (in GB), and the hidden and intermediate size for four different-sized models when applying DELTAMIX.

| Model Size | Optimization | Memory Usage | Hidden (Intermediate) Size |
|---|---|---|---|
| DeepSeek-R1-Distill-Qwen-1.5B | 16.90 | 4.76 | 1536(8960) |
| DeepSeek-R1-Distill-Qwen-7B | 59.38 | 12.62 | 3584(18944) |
| DeepSeek-R1-Distill-Qwen-14B | 71.87 | 16.23 | 5120(13824) |
| DeepSeek-R1-Distill-Llama-70B | 171.80 | 47.53 | 8192(28672) |

**ILP Accelerate.** We solve the ILP using open-source solvers in our paper. However, this process can be accelerated by 6x times if we switch from SCIP to proprietary solvers such as COPT (Ge et al., 2023) when handling ILP problems. Given that more than half of the quantization time for DELTAMIX is dominated by the ILP-solving process, adopting such commercial solvers could significantly enhance DELTAMIX's efficiency in practice. Additionally, DELTAMIX can be further optimized through other means, such as by limiting the number of candidate bitwidths (e.g., from 8 to 4).

E.5    PERFORMANCE UNDER DIFFERENT COMPRESSION RATIO

To show that DELTAMIX can apply to arbitrary compression ratios, we evaluated DeepSeek-R1-Distill-Qwen-7B and Qwen2.5-Math-7B-Instruct at four compression ratios, as shown in Table 18. The performance of DELTAMIX decreases as the compression ratio increases. This is expected,

Table 18: Performance of DELTAMIX under different compression ratios $1/\alpha$.

| $\alpha$ | DeepSeek-R1-Distill-Qwen | | Qwen2.5-Math-Instruct | | Average |
|---|---|---|---|---|---|
| | Math500 | AIME2024 | Math500 | GSM8K | |
| 3/16 | 86.4 | 36.7 | 77.2 | 95.6 | 74.0 |
| 2/16 | 85.8 | 33.3 | 77.4 | 95.1 | 72.9 |
| 1/16 | 83.2 | 33.3 | 77.6 | 94.8 | 72.2 |
| 1/32 | 76.8 | 26.7 | 73.4 | 91.6 | 67.1 |

Table 19: The detailed storage overhead between DELTAMIX and Delta-CoMe.

| | DELTAMIX | | | Delta-CoMe | | |
|---|---|---|---|---|---|---|
| | Quantized Weights | Other Parameters | Total Storage | Quantized Weights | Other Parameters | Total Storage |
| Qwen2.5-Coder-7B-Instruct | 0.81 | 0.06 | 0.87 | 0.81 | 0.06 | 0.87 |
| Qwen2.5-VL-7B-Instruct | 0.81 | 0.06 | 0.87 | 0.81 | 0.06 | 0.87 |
| Qwen2.5-Math-7B-Instruct | 0.81 | 0.06 | 0.87 | 0.81 | 0.06 | 0.87 |
| DeepSeek-R1-Distill-Qwen-7B | 0.81 | 0.05 | 0.86 | 0.81 | 0.06 | 0.87 |
| MetaMath-13B-V1.0 | 1.51 | 0.09 | 1.60 | 1.51 | 0.12 | 1.63 |
| Qwen2.5-Coder-14B-Instruct | 1.63 | 0.11 | 1.74 | 1.63 | 0.12 | 1.75 |
| DeepSeek-R1-Distill-Qwen-14B | 1.63 | 0.10 | 1.73 | 1.63 | 0.12 | 1.75 |
| llava-v1.5-13b | 1.51 | 0.10 | 1.61 | 1.51 | 0.12 | 1.63 |

as a higher compression ratio indicates a reduced capacity of the quantized model to preserve information from the original model. Notably, baselines like BitDelta and Delta-CoMe cannot apply to other compression ratios except $\alpha = 1/16$. BitDelta quantizes $\Delta\mathbf{W}$ to a fixed 1 bit, resulting in a constant compression ratio. For Delta-CoMe, the empirically determined mixed-precision scheme is fixed and does not offer a clear method for allocating mixed precision at other compression ratios. In contrast, DELTAMIX enables the compression of $\Delta\mathbf{W}$ to arbitrary ratios, offering greater flexibility and broader applicability.

### E.6 BUDGET PARITY BETWEEN DELTAMIX AND DELTA-COME

To more accurately compare the storage overhead of DELTAMIX with the strongest baseline, Delta-CoMe, and to demonstrate the fairness of the experiment, we compare the storage overhead of models in our main experiments. We divide the total storage into two components: "Quantized Weights" representing the storage used by quantized parameters, and "Other Parameters" include non-weight parameters such as Scales (stored in 16 bits) and Zeros (stored according to their quantization bitwidth. Table 19 demonstrates that DELTAMIX exhibits lower storage overhead compared with Delta-CoMe. This trend is further illustrated in Figure 3, where Delta-CoMe supports up to 8 models, whereas DELTAMIX can deploy 12 simultaneously. These results clearly demonstrate the superior efficiency of DELTAMIX.

### E.7 ANALYZING THE BIT ALLOCATION RESULTS

We investigate the bit allocation results across different weight types and layers using the Qwen2.5-Math-7B-Instruct model. Figure 4 shows the memory allocated for each bit-width. Overall, the bit allocation results for different weight types and layers are different. The V_Proj, K_Proj and O_proj in the self-attention layer exhibit a similar allocation trend. For the other four weight types, the bit allocation results differ. For instance, Down_Proj allocates more 2-bit units at the beginning compared to other weight types.

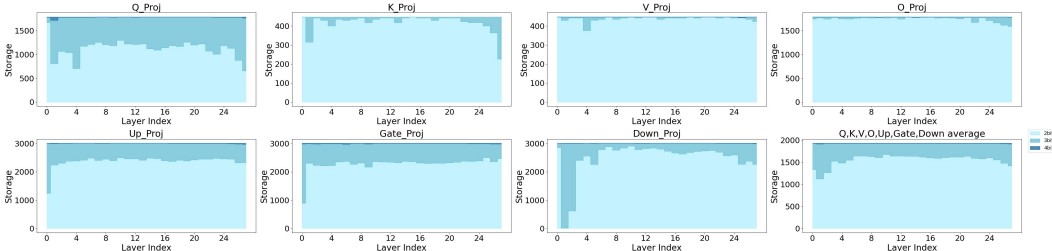

Figure 4: GPU memory usage with quantization bits across layers of Qwen2.5-Math-7B-Instruct.

Delta-CoMe (Ping et al., 2024) empirically posits that singular vectors corresponding to larger singular values are more significant and, therefore, necessitate higher-bit representations. We further examine whether DELTAMIX adheres to this assumption, specifically by using singular values alone to evaluate importance. We compute the Kendall rank correlation coefficient $\tau$, between the bit sequence and the singular value sequence for each $\mathbf{W}$. The coefficient is a measure of rank correlation, ranging from -1 to 1, reflecting the similarity of the orderings of the data when ranked by each of the quantities. If the method strictly adhered to the assumption of using singular values alone for importance assessment, singular vectors with larger singular values would always receive higher bit-width, resulting in a consistent $\tau = 1$ across all $\mathbf{W}$. However, for the DeepSeek-R1-Distill-Qwen-7B

model with DELTAMIX, we observe a $\tau$ of 0.95 for the $\mathbf{W}$ of the key projection at layer 28. This indicates that DELTAMIX goes beyond singular values, taking into account both the "scaling" term and the "difference" term.

## E.8 ANALYZING THE QUANTIZATION ERROR ACROSS WEIGHT TYPES AND LAYERS

Table 20: Average quantization error ($\times$ 1e2) accross different type of linears with Eq. (1)."Low", "Mid", and "High" denote the first 9 layers, layers 9 to 17, and the last 10 layers, respectively. "All" and "Out" denote the average error across all activations and the average error of the top 1% of activations.

**Q_proj**

| Layer | Low | | Mid | | High | |
|---|---|---|---|---|---|---|
| Type | All | Out | All | Out | All | Out |
| Low-Rank | 0.26 | 0.32 | 0.54 | 0.76 | 1.33 | 1.64 |
| BitDelta | 0.18 | 0.37 | 0.27 | 0.37 | 0.68 | 1.00 |
| Delta-CoMe | 0.13 | 0.14 | 0.32 | 0.41 | 0.81 | 0.91 |
| DELTAMIX | **0.10** | **0.11** | **0.25** | **0.32** | **0.64** | **0.73** |

**K_proj**

| Layer | Low | | Mid | | High | |
|---|---|---|---|---|---|---|
| Type | All | Out | All | Out | All | Out |
| Low-Rank | 0.06 | 0.07 | 0.11 | 0.13 | 0.19 | 0.29 |
| BitDelta | **0.03** | **0.03** | **0.05** | **0.06** | **0.08** | **0.12** |
| Delta-CoMe | **0.03** | **0.03** | 0.06 | 0.07 | 0.12 | 0.21 |
| DELTAMIX | **0.03** | **0.03** | **0.05** | 0.07 | 0.10 | 0.18 |

**V_proj**

| Layer | Low | | Mid | | High | |
|---|---|---|---|---|---|---|
| Type | All | Out | All | Out | All | Out |
| Low-Rank | 0.03 | 0.03 | 0.06 | 0.08 | 0.39 | 1.11 |
| BitDelta | **0.01** | **0.01** | **0.03** | **0.03** | **0.18** | 0.69 |
| Delta-CoMe | 0.02 | 0.02 | 0.04 | 0.05 | 0.24 | 0.85 |
| DELTAMIX | 0.02 | 0.02 | 0.04 | 0.05 | 0.21 | **0.67** |

**O_proj**

| Layer | Low | | Mid | | High | |
|---|---|---|---|---|---|---|
| Type | All | Out | All | Out | All | Out |
| Low-Rank | 0.23 | 0.40 | 0.70 | 1.54 | 8.52 | 69.00 |
| BitDelta | 0.10 | 0.14 | **0.28** | 0.46 | 10.44 | 895.98 |
| Delta-CoMe | 0.08 | 0.13 | 0.32 | 0.47 | 3.53 | 17.02 |
| DELTAMIX | **0.07** | **0.12** | 0.30 | **0.45** | **3.18** | 22.31 |

**Up_proj**

| Layer | Low | | Mid | | High | |
|---|---|---|---|---|---|---|
| Type | All | Out | All | Out | All | Out |
| Low-Rank | 4.78 | 4.50 | 2.67 | 3.18 | 13.70 | 14.95 |
| BitDelta | 4.71 | 3.85 | **1.19** | **1.32** | 13.30 | 11.61 |
| Delta-CoMe | 2.10 | 2.08 | 1.60 | 1.90 | 7.67 | 9.37 |
| DELTAMIX | **1.83** | **1.74** | 1.36 | 1.59 | **6.58** | **8.89** |

**Gate_proj**

| Layer | Low | | Mid | | High | |
|---|---|---|---|---|---|---|
| Type | All | Out | All | Out | All | Out |
| Low-Rank | 6.35 | 3.85 | 3.16 | 0.72 | 13.53 | 4.02 |
| BitDelta | 9.01 | 4.47 | 1.60 | 0.65 | 10.32 | 5.87 |
| Delta-CoMe | 2.64 | 2.90 | 1.88 | 0.84 | 7.73 | 3.02 |
| DELTAMIX | **2.28** | **2.22** | **1.57** | **0.59** | **6.65** | 2.07 |

**Down_proj**

| Layer | Low | | Mid | | High | |
|---|---|---|---|---|---|---|
| Type | All | Out | All | Out | All | Out |
| Low-Rank | 1.05 | 5.52 | 3.28 | 4.94 | 110.20 | 7470.34 |
| BitDelta | 1.21 | 2.35 | 0.87 | 1.45 | 115.60 | 11735.05 |
| Delta-CoMe | 0.33 | 1.86 | 1.05 | 1.57 | 32.66 | 1851.91 |
| DELTAMIX | **0.31** | **1.62** | **1.02** | **1.43** | **30.30** | **1669.95** |

**Average**

| Layer | Low | | Mid | | High | |
|---|---|---|---|---|---|---|
| Type | All | Out | All | Out | All | Out |
| Low-Rank | 1.82 | 3.67 | 1.50 | 2.84 | 21.12 | 1890.34 |
| BitDelta | 2.18 | 2.81 | 0.61 | 1.08 | 21.51 | 3162.58 |
| Delta-CoMe | 0.76 | 1.79 | 0.75 | 1.33 | 7.54 | 470.82 |
| DELTAMIX | **0.66** | **1.46** | **0.66** | **1.12** | **6.81** | **426.20** |

