# OpenReview forum: "Enhancing Delta Compression in LLMs via SVD-based Quantization Error Minimization"
_ICLR.cc/2026/Conference — Submitted to ICLR 2026_

### Official Review · Reviewer_g1TA · 2025-10-25

**Soundness:** 3
**Presentation:** 2
**Contribution:** 2
**Rating:** 4
**Confidence:** 4

**Summary:**

This work proposes an SVD-guided approach for low-bit mixed-precision compression of weight deltas (differences between original and fine-tuned models). The authors first provide a theoretical derivation showing that mixed-precision quantization is beneficial for the V matrix but less useful for the U matrix in SVD decomposition. To compensate for the error incurred by quantization of the V matrix during the U quantization step, the authors introduce Reconstruction Target Correction, which shifts U to a new optimal value. The analysis is accompanied by empirical evidence. Based on this foundation, the authors formulate the optimal compression as an integer programming problem and leverage a dedicated solver. The proposed method is validated on several large language and vision models and compared with prior work on delta compression.

**Strengths:**

* The SVD-guided search for mixed-precision quantization appears to be novel in the context of model compression.
* Delta-Mix noticeably outperforms baselines in terms of final accuracy for the same compression target.

**Weaknesses:**

* While theoretically and practically sound, the proposed method—formulated as compression of a model delta of the same size as the original model—still seems less appealing than established PEFT techniques [1, 2, 3, 4]. LoRA adapters can be one or two orders of magnitude smaller than the total number of model parameters, yet remain competitive with full fine-tuning when properly tuned [5]. The learning rate adopted in the experiments (4e-5) may not be optimal for LoRA.

---
References

[1] Hu, Edward J., et al. "Lora: Low-rank adaptation of large language models." ICLR 1.2 (2022): 3.

[2] Liu, Shih-Yang, et al. "Dora: Weight-decomposed low-rank adaptation." Forty-first International Conference on Machine Learning. 2024.

[3] Zhang, Qingru, et al. "Adalora: Adaptive budget allocation for parameter-efficient fine-tuning." arXiv preprint arXiv:2303.10512 (2023).

[4] Kopiczko, Dawid J., Tijmen Blankevoort, and Yuki M. Asano. "Vera: Vector-based random matrix adaptation." arXiv preprint
arXiv:2310.11454 (2023).

[5] https://thinkingmachines.ai/blog/lora/

**Questions:**

* Can the proposed method be applied to LoRA adapters? This would potentially enable even higher compression rates and the possibility of serving a large number of fine-tuned versions of a given model simultaneously.


* The proposed method seems to be quantized representation-agnostic. Can Delta-Mix be combined with a vector quantization scheme [1, 2, 3] to achieve even higher compression rates with minimal performance degradation?


* LoRA may sometimes lack sufficient expressiveness to fully capture the difference between two models when the difference is substantial. How well do sparse + low-rank adapters [4] perform in this context?

---
References

[1] Van Baalen, Mart, et al. "Gptvq: The blessing of dimensionality for llm quantization." arXiv preprint arXiv:2402.15319 (2024).

[2] Egiazarian, Vage, et al. "Extreme compression of large language models via additive quantization." arXiv preprint arXiv:2401.06118 (2024).

[3] Chee, Jerry, et al. "Quip: 2-bit quantization of large language models with guarantees." Advances in Neural Information Processing Systems 36 (2023): 4396-4429.

[4] Nikdan, Mahdi, et al. "Rosa: Accurate parameter-efficient fine-tuning via robust adaptation." arXiv preprint arXiv:2401.04679 (2024).

---

> ### Author Response · Authors · 2025-11-25
> **Official Comment by Authors - Part 1**
>
> # Q1: The learning rate adopted in the experiments may not be optimal for LoRA.
>
> A1: Thank you for raising this concern. **Our learning rate for LoRA training indeed lies within the optimal range reported in [1].** LoRA involves two hyperparameters: the rank $r$ and the scale factor $\alpha$, with the parameterization $W' = W + \frac{\alpha}{r}BA$. According to [1], all experiments adopt $\alpha = 32$, and Figure 2 indicates an optimal learning-rate range of $lr^{*} \in [10^{-4}, 10^{-3})$.
>
> In this work, LoRA is trained using the LLaMA-Factory framework [2], which sets $\alpha = 2r$ by default. To achieve a compression ratio of $1/16$, we use $r = 128$, yielding $\alpha = 256$, which is 8x larger than the value in [1]. Because the gradients of $A$ and $B$ scale linearly with $\alpha$, **an increase in $\alpha$ effectively rescales the learning rate.** Although we set $lr = 4\times10^{-5}$, **considering our $\alpha$ is 8 times that in [1]**, the corresponding effective learning rate is $lr = 4\times10^{-5} \times 8 = 3.2\times10^{-4}$, which falls within the optimal interval reported in [1].
>
> # Q2: While theoretically and practically sound, the proposed method—formulated as compression of a model delta of the same size as the original model—still seems less appealing than established PEFT techniques
>
> A2: Thank you for your question. We believe that **DeltaMix method demonstrates a strong appeal:**
>
> - **Compatibility with PEFT methods:** As noted in our response to A3, DeltaMix can be applied not only in full fine-tuning settings but also in PEFT scenarios. In particular, **integrating DeltaMix with LoRA further enhances LoRA’s 4× compression ratio while maintaining virtually the same performance after compression.**
>
> - **Full fine-tuning methods are essential:** Significant research has been conducted, and fine-tuned models have been released based on the principle of full fine-tuning [3] [4], including the aligned LLMs referenced in our paper. As long as full fine-tuning remains widely used, **DeltaMix presents an appealing approach for compressing the incremental parameter $\Delta \mathbf{W}$, thereby reducing both memory consumption and computational cost.**
>
> - **The fully fine-tuned model with DeltaMix consistently surpasses LoRA, even under high compression.** In A3, at a compression ratio of $\alpha = \frac{1}{64}$, the compressed fully fine-tuned model achieves a higher score (39.1) than the LoRA model at the same ratio (35.4). These results indicate that integrating DeltaMix with a fully fine-tuned model yields superior performance compared to using LoRA alone or LoRA combined with DeltaMix.
>
> # Q3: Can the proposed method be applied to LoRA adapters? This would potentially enable even higher compression rates and the possibility of serving a large number of fine-tuned versions of a given model simultaneously.
>
> A3: Thank you for your insightful question. Indeed, **DeltaMix can be applied to LoRA to further enhance the compression ratio.** To validate the feasibility of this approach, we further compressed the model obtained in Section 5.2, after both LoRA training and full fine-tuning, to a compression ratio of $\alpha = \frac{1}{64}$.
>
> The results show as follows:
>
> | Method| $\alpha$ | Humaneval | Mbpp | Math500 | GSM8K | AVERAGE |
> | - | - | - | - | - | - | - |
> | LoRA| 1/16 | 34.1| 47.7 | 9.4 | 50.9| 35.5|
> | DeltaMix| 1/16 | 43.3| 50.2 | 13.5| 56.1| 40.8|
> | DeltaMix-LoRA | 1/64 | 34.1| 47.6 | 10.4| 49.6| 35.4|
> | DeltaMix| 1/64 | **39.0** | **51.6** | **11.6**| **54.3**| **39.1** |
>
> - The results show that **the performance of LoRA compressed to $\alpha = \frac{1}{64}$ remains nearly identical to the original LoRA model (35.4 vs. 35.5).** This finding demonstrates that applying DeltaMix to LoRA effectively achieves additional compression without significant performance loss, thus offering substantial storage savings.
>
> - Notably, **Baselines like BitDelta and Delta-CoMe in the baseline cannot apply to LoRA.** BitDelta directly quantizes $\Delta \mathbf{W}$ to 1 bit without employing any low-rank approximation. Consequently, it cannot effectively utilize the low-rank properties inherent in LoRA. For Delta-CoMe, the empirically determined mixed-precision scheme is fixed and does not offer a clear method for allocating mixed precision at other compression ratios. In contrast, **DeltaMix allows compression of $\Delta \mathbf{W}$ to arbitrary ratios**, making it more flexible and practically advantageous.
>
> [1] https://thinkingmachines.ai/blog/lora/
>
> [2] https://github.com/hiyouga/LLaMA-Factory
>
> [3] Language Models are Super Mario: Absorbing Abilities from Homologous Models as a Free Lunch. ICML 2024
>
> [4] DeepSeek-R1: Incentivizing Reasoning Capability in LLMs via Reinforcement Learning. arXiv:2501.12948v1

---

> ### Author Response · Authors · 2025-11-25
> **Official Comment by Authors - Part 2**
>
> # Q4: Can Delta-Mix be combined with a vector quantization scheme to achieve even higher compression rates with minimal performance degradation?
>
> A4: Thank you for your question. **DeltaMix is indeed quantization representation-agnostic and can be applied to the three methods you mentioned.** However, compared with GPTQ and Quip [1], the approaches GPTQV [2] and AQLM [3] present the following limitations:
>
> * **Slower quantization speed.** GPTQ and Quip do not require additional training. Quip employs two random orthogonal matrices to address parameter coherence without any optimization steps. In contrast, GPTQV and AQLM necessitate additional training of codebooks, which increases the computational time required during simulation.
>
> * **Reduced simulation accuracy.** Because GPTQV and AQLM share a codebook across parameter rows, they cannot support per-channel quantization as in GPTQ. Consequently, quantization across rows cannot remain fully independent, making it difficult to accurately estimate per-row (rank-wise) quantization error at any bit width. This shared-codebook design may introduce bias in the simulated quantization error across different bit levels, thereby compromising the optimality of subsequent Integer Linear Programming (ILP) solutions.
>
> In summary, DeltaMix can be applied to all three methods; however, we recommend using it with approaches that allow independent row-wise quantization (e.g., per-channel quantization in GPTQ) to enhance simulation accuracy.
>
> # Q5: LoRA may sometimes lack sufficient expressiveness to fully capture the difference between two models when the difference is substantial. How well do sparse + low-rank adapters perform in this context?
>
> A5:  Thank you for your insightful suggestions. We have found that RoSA indeed outperforms standard LoRA on more challenging tasks, thereby enhancing LoRA’s expressive capability (see Table 1 in [4]).
>
> **Integrating sparse + low-rank approaches such as RoSA with DeltaMix is a promising direction to explore**; It lies beyond the scope of this study: based on minimizing quantization errors, proving the necessity of mixed-precision quantization in SVD-based methods, and deriving the optimal mixed-precision scheme. We plan to investigate the combination of DeltaMix with approaches like RoSA in future work.
>
> [1] Chee, Jerry, et al. "Quip: 2-bit quantization of large language models with guarantees." Advances in Neural Information Processing Systems 36 (2023): 4396-4429.
>
> [2] Van Baalen, Mart, et al. "Gptvq: The blessing of dimensionality for llm quantization." arXiv preprint arXiv:2402.15319 (2024).
>
> [3] Egiazarian, Vage, et al. "Extreme compression of large language models via additive quantization." arXiv preprint arXiv:2401.06118 (2024).
>
> [4] Nikdan, Mahdi, et al. "Rosa: Accurate parameter-efficient fine-tuning via robust adaptation." arXiv preprint arXiv:2401.04679 (2024).

---

> > ### Comment · Reviewer_g1TA · 2025-11-26
> >
> > Thank you for the response. After reading the rebuttals addressed to me and other reviewers, I decided to raise my score.

---

> > > ### Author Response · Authors · 2025-11-26
> > >
> > > Thank you for raising the score! We are glad that the clarifications and additional experiments addressed your concerns. This serves as an important encouragement and recognition of our work.

---

### Official Review · Reviewer_mFqR · 2025-10-31

**Soundness:** 3
**Presentation:** 3
**Contribution:** 3
**Rating:** 4
**Confidence:** 4

**Summary:**

This paper presents DeltaMix, an adaptive mixed-precision delta-compression framework for fine-tuned large language models.
The method decomposes delta weights using SVD and formulates quantization as minimizing layer-wise reconstruction error.
Unlike prior empirical approaches, DeltaMix derives a mathematically grounded formulation showing why mixed-precision is necessary and models bit allocation as a 0/1 integer-linear program with an additional RTC step.
Extensive experiments on reasoning, math, code, and multimodal benchmarks across diverse models demonstrate consistent performance gains, while also reducing GPU memory usage.

**Strengths:**

**1. Strong theoretical foundation.** The work formalizes SVD-based delta-compression as an explicit quantization-error-minimization problem and proves the necessity of mixed-precision allocation, advancing the theoretical rigor of delta-compression research.

**2. Comprehensive empirical validation.** Evaluations on 7B and 14B LLMs across four domains (reasoning, math, code, vision-language) show clear and reproducible gains over Delta-CoMe, BitDelta, and low-rank baselines.

**3. Practical deployment benefits with thorough system analysis.** The paper provides valuable end-to-end evaluation showing 6× memory savings and superior scaling properties, enabling deployment of different models for uncompressed approaches.
The analysis of prefill time, generation speed, and varying arrival rates demonstrates real-world applicability beyond just accuracy metrics.

**Weaknesses:**

**1. Limited scalability analysis.** While integer-linear optimization is solved once per model, reported solving times (≈ 30 min for 7B) may become impractical for larger or frequent model updates.
Discussion on scaling to 70B+ models is missing.

**2. Ablation study.** Although four task types are covered, the paper lacks ablation on calibration-set size, bit-budget sensitivity, or robustness under distribution shift, which are important for real-world deployment.

**3. Computational overhead.** Table 10 shows DeltaMix requires 3× more time than Delta-CoMe (per block), totaling 1-3 hours for full models.
While the paper dismisses this as "acceptable since quantization is performed only once," this represents significant overhead for practitioners, especially for larger models or when iterating on model development.

**Questions:**

**1. Complexity and scalability.** How does the integer-program’s solving time and memory footprint scale with layer size and number of candidate bit-widths?
Could approximate or heuristic solvers yield near-optimal results faster?

**2. Sensitivity towards calibration data.** How robust is the bit-allocation when the calibration set is small or domain-mismatched?
Does performance degrade significantly with limited calibration data, and how does this compare to baselines that may be less calibration-dependent?

**3. Generalization to other compression forms.** Could the same error-minimization principle be adapted for pruning or hybrid pruning-quantization pipelines?

**Justification for Rating.**

The paper presents a novel and theoretically motivated approach to delta compression.
However, the experimental section lacks sufficient analysis of scalability, sensitivity, and efficiency trade-offs, which limits the practical completeness of the proposed framework.
I am open to raising the score if these concerns are adequately addressed in the rebuttal.

---

> ### Author Response · Authors · 2025-11-25
> **Official Comment by Authors - Part 1**
>
> # Q1: Limited scalability analysis. While integer-linear optimization is solved once per model, reported solving times (≈ 30 min for 7B) may become impractical for larger or frequent model updates. Discussion on scaling to 70B+ models is missing.
>
> A1: Thanks for your question. To demonstrate that DeltaMix is practical for larger models, we conducted experiments on quantizing a 70B model (DeepSeek-R1-Distill-Llama-70B). The process took only 9.26 hours and utilized 47.5GB of GPU memory. This computational cost is acceptable considering the following factors:
>
> - **Time portion in the training cycle**:  When developing a model for a downstream task, we integrate DeltaMix into the training cycle after full fine-tuning. **The computational overhead introduced by DeltaMix is minimal compared to that of the training cycle.** For example, as described in Appendix D.2, we fine-tuned DeepSeek-LLM-7B-Base on 50K samples for three epochs using eight L40 GPUs, which took about 5 hours in total (equivalent to roughly 40 GPU-hours on a single device). Under this configuration, DeltaMix needs 1.2h to quantize 7B models, accounting for only a small portion of the training cycle, approximately $2.9$% .
>
> - **Time portion in the deployment cycle**: **The time required for DeltaMix remains negligible when compared to the deployment cycle of the models,** as DeltaMix typically requires a few hours, while the deployment typically spans months or years [1].
>
> - In Lines 956-962, we introduce methods to accelerate the solution of the ILP problem, such as changing to a faster solver [2] and using smaller candidate bit-widths, which can improve the speed of ILP problems by at least 6x [3]. In A4, we further reduce the quantization time by implementing a more efficient code for the simulation process. **Therefore, we believe our method is fully practical for larger or frequent model updates.**
>
>
> # Q2: Ablation study. Although four task types are covered, the paper lacks ablation on calibration-set size, bit-budget sensitivity, or robustness under distribution shift, which are important for real-world deployment.
>
> A2:
> - Based on your suggestion, we conducted experiments in **A5** on the domains and sizes of the calibration dataset (**lines 391-401**). **DeltaMix demonstrate robustness under these conditions.**
>
> - For ablation on bit-budget, we conducted experiments in **Section E.5 (lines 963-988)** by varying the compression ratios from 16/3 to 32. The performance of DeltaMix decreases as the compression ratio increases. Notably, baselines like BitDelta and Delta-CoMe cannot apply to other compression ratios except $\alpha=$1/16. BitDelta quantizes $\Delta \mathbf{W}$ to a fixed 1 bit, resulting in a constant compression ratio. For Delta-CoMe, the empirically determined mixed-precision scheme is fixed and does not offer a clear method for allocating mixed precision at other compression ratios. **In contrast, DeltaMix enables the compression of $\Delta \mathbf{W}$ to arbitrary ratios, offering greater flexibility and broader applicability.**
>
> # Q3: Computational overhead. Table 10 shows DeltaMix requires 3× more time than Delta-CoMe (per block), totaling 1-3 hours for full models. While the paper dismisses this as "acceptable since quantization is performed only once," this represents significant overhead for practitioners, especially for larger models or when iterating on model development.
>
> A3: Thanks for your question. Please refer to our response A1.
>
> [1] https://help.openai.com/en/articles/6825453-chatgpt-release-notes#h_34439eda3a
>
> [2] https://www.shanshu.ai/copt/
>
> [3] https://plato.asu.edu/ftp/path.html

---

> ### Author Response · Authors · 2025-11-25
> **Official Comment by Authors - Part 2**
>
> # Q4: Complexity and scalability. How does the integer-program’s solving time and memory footprint scale with layer size and number of candidate bit-widths? Could approximate or heuristic solvers yield near-optimal results faster?
> A4: Thanks for your question. We  address this concern in **lines 925-962**:
> - **Solving time with various numbers of candidate bit-widths**: Because the ILP solver operates on the CPU, **the GPU memory footprint remains unchanged regardless of the number of candidate bit widths.** Here, we report the latencies applying DeltaMix to quantize a single transformer block of Qwen2.5-Math-7B-Instruct in three parts:
>
>   * **Simulation**: The time cost of estimating quantization loss for all candidate bit-widths with a sequential algorithm. **This is what we adopt in the paper's implementation.**
>   * **Simulation(Parallel)**: Simulation time cost with a parallel algorithm.
>   * **Optimization**: The time cost for solving ILP in one transformer block.
>   * **Quantization**: The time cost of quantizing one transformer block according to the quantization scheme.
>
>   The results indicate that the **“Simulation” with a parallel algorithm and “Quantization” remain nearly constant.** For simulations using a parallel algorithm, the losses corresponding to different bit-widths can be computed concurrently on a GPU, achieving a speedup of 13 seconds per block compared with the sequential algorithm. For Quantization, the parameters of any rank, the assigned bit width is fixed once the mixed-precision scheme is determined; thus, varying the range of candidate bit widths has a negligible effect.
>
>   In contrast, the **Optimization stage is directly affected by the number of available bit widths and therefore dominates the overall runtime**. The bit-width range determines the size of the ILP solution space, leading to substantial increases in optimization time. For instance, expanding the bit-width set from 3 to 8 doubles the runtime (30.8 vs. 63.0).  Consequently, larger bit-width sets result in significantly longer optimization times.
>   For time-constrained quantization tasks, the bit-width set can be reduced without degrading accuracy. Our quantization results show that most chosen bit widths fall within $\{0, 2, 3, 4\}$. **Restricting candidates to these four values cuts the quantization time by nearly half compared with the original(8 bit-widths) setting (35.0 vs. 63.0)**.
>
> | #Candidate Bit-widths | Simulation | Simulation(Parallel) | Optimization | Quantization | Total Time(w/o Parallel) | Total Time(w Parallel) |
> | - | - | - | - | - | - | - |
> | 3 | 22.29| 25.67| 30.80| 18.39| 71.48| 74.86|
> | 4 | 26.05| 26.51| 35.01| 21.42| 82.48| 82.94|
> | 5 | 28.41| 25.76| 39.68| 19.78| 87.87| 85.22|
> | 6 | 32.18| 27.14| 46.40| 19.88| 98.46| 93.42|
> | 7 | 37.46| 26.53| 55.14| 20.71| 113.31 | 102.38 |
> | 8 | 39.66| 25.96| 63.00| 20.71| 123.37 | 109.67 |
>
>
> - **Memory and solving time scale with layer size**: We measured the total time and memory needed to solve the ILP for a single transformer block using four different-sized models in three parts:
>   - **Optimization:** The time cost of estimating quantization loss under a candidate bit-width configuration (in seconds).
>   - **Memory Usage:** Memory consumption of DeltaMix during quantization for these four models (in GB).
>   - **Hidden (Intermediate) Size:** Each model’s hidden and intermediate dimensions.
>
>   The results demonstrate that ILP runtime is primarily determined by the hidden dimension of an individual linear layer. **It should also be noted that, to satisfy open-source requirements, our experiments employed a slower open-source solver (SCIP)**. In practice, the use of faster commercial ILP solvers or a reduced set of candidate bit-widths can substantially accelerate ILP solving. **Thus, the reported optimization times should be interpreted as a lower bound for real-world deployments.**
>
>   Regarding memory usage, DeltaMix, as a per-layer quantization method, exhibits modest GPU memory requirements. Quantizing a 70B-parameter model fits within a single L20 GPU, indicating that **DeltaMix is not resource-intensive** and is therefore suitable for large-scale applications in resource-constrained environments or for parallel quantization of multiple models.
>
> | Model |Optimization| Memory Usage | Hidden (Intermediate) Size |
> | - | - | - | - |
> | DeepSeek-R1-Distill-Qwen-1.5B| 16.9| 4.8| 1536(8960) |
> | DeepSeek-R1-Distill-Qwen-7B | 59.4| 12.6 | 3584(18944) |
> | DeepSeek-R1-Distill-Qwen-14B| 71.9| 16.2 | 5120(13824)|
> | DeepSeek-R1-Distill-Llama-70B| 171.8 | 47.5 | 8192(28672)|
>
> - **Further reducing the time cost with a commercial ILP solver**: In lines 956-962, we introduce methods to accelerate the solution of ILP problems, such as switching to a faster solver [1] and using a smaller candidate bit width. This can improve the speed of ILP problems by at least 6x [2].
>
> [1] https://www.shanshu.ai/copt/
>
> [2] https://plato.asu.edu/ftp/path.html

---

> ### Author Response · Authors · 2025-11-25
> **Official Comment by Authors - Part 3**
>
> # Q5: Sensitivity towards calibration data. How robust is the bit-allocation when the calibration set is small or domain-mismatched? Does performance degrade significantly with limited calibration data, and how does this compare to baselines that may be less calibration-dependent?
>
> A5: Thanks for your question. We add the experiments in Section 4.4 (**line 391-401**). Following Delta-CoMe[4], we randomly selected 128 data points of length 2048 from the C4 dataset as the calibration dataset for all experiments.
>
> - **Domain of calibration set**: We quantized the Qwen2.5-Math-7B-Instruct model using calibration data drawn from three different datasets. Each calibration set contains 128 randomly sampled sequences of length 2048. Because MetaMathQA does not contain enough sequences of this length, we oncatenated multiple question–answer pairs using a few-shot format.
> **The results show that DeltaMix performs well on all calibration setups, confirming DeltaMix’s robustness to the domain variations in the calibration dataset.**
> | Dataset | Math500 | GSM8K | Average |
> | - | - | - | - |
> | C4| **77.6**| **94.8**| **86.2**|
> | Wikitext2 | 76.6| **94.8**| 85.7|
> | MetaMathQA| 75.4| 93.6| 84.5|
>
> - **Size of calibration set**: To assess DeltaMix’s sensitivity to calibration-set size, we conduct experiments with Qwen2.5-Math-7B-Instruct using calibration sets of 16, 32, 64, 128, and 256 samples from C4.
>
>   The results show that even with only 16 calibration samples, performance remains close to the optimal setting of 128 samples, demonstrating that **DeltaMix maintains strong performance with small calibration sets.**
>
> | Sample Size | Math500  |GSM8K| Average|
> | - | - | - | - |
> | 16 | 76.4 | 94.5| 85.5|
> | 32| 76.2| **95.1** | 85.7 |
> | 64 | 76.8 | 94.3 | 85.6 |
> | 128 | **77.6** | 94.8 | **86.2** |
> | 256 | 76.0| 94.1| 85.1|
>
> - **Compare to a calibration-independent baseline**: In Tables 1 and 2, we compare DeltaMix with a naive SVD-based truncation baseline (the “Low-Rank’’ rows).  Across eight models and four downstream tasks, the results show that **DeltaMix yields substantially higher average performance than simple SVD truncation**.
>
> # Q6: Generalization to other compression forms. Could the same error-minimization principle be adapted for pruning or hybrid pruning-quantization pipelines?
>
> A6: Thank you for your insightful question. **Our method can also be incorporated into SVD-based pruning or hybrid pruning–quantization pipelines**. For any SVD-based compression method, if the loss function can be modeled using Equation (3), then the corresponding loss function in Equation (5) necessarily exists. Consequently, the losses associated with different ranks under varying sparsity rates (or other hyperparameters) can be computed through simulation. The optimal sparsity rate (or hyperparameter) allocation scheme is then determined by solving the ILP described in Section 3.2.

---

> ### Comment · Reviewer_mFqR · 2025-11-25
>
> Thank you for the detailed rebuttal.
> However, I am concerned on two parts for the initial rebuttal: 1) clarified distinction between answers of a total of 6 weeknesses and questions, and 2) the delayed update of the manuscript.
> Would you mind to clarify your answers of each points thoroughly under a top-down manner and update the manuscript to contain all additionally provided information?

---

> ### Author Response · Authors · 2025-11-26
>
> Thank you for your valuable suggestions. Based on your advice, we have reorganized the weaknesses and questions, answering them in order, as well as updated our manuscript with the additionally provided information. We hope our latest response can address your concerns. If you have any further questions or comments regarding our paper, please feel free to let us know. We will address them as soon as possible.

---

> > ### Comment · Reviewer_mFqR · 2025-11-28
> >
> > Thanks for the rebuttal.
> > However, I am not still convinced about its 1) scalability due to the running time, and 2) generalizability towards other compression methods.
> > Thus, I continue to lean toward rejecting the manuscript in its present form.

---

> ### Author Response · Authors · 2025-11-28
>
> Thank you for your feedback. Regarding the concerns you still remain, we would like to provide further explanation and clarification below:
>
> # Q1: Scalability due to the running time
> A1: We understand your concern; however, we would like to emphasize that DeltaMix is indeed scalable.
> - **Resource Efficiency**: As demonstrated in A1, A4, and line 911-955, DeltaMix can quantize models ranging from 1.5B to 70B in just 0.27 to 9.26 hours using a single L40 GPU. This illustrates that DeltaMix is **not resource-intensive**, requiring only one GPU and minimal time within the overall model development cycle. For instance, in the finetuning-quantization pipeline, the quantization process with DeltaMix accounts for only **2.9% of the total GPU hours** when applied to the DeepSeek-LLM-7B-Base with a 150k training dataset.
> - **Potential for Further Acceleration**: DeltaMix can be further **accelerated by over 2x**. As indicated in A1, line 955-962, the ILP solving process can be sped up by at least 6x by switching from our open-sourced SCIP to the commercial COPT. Since the ILP-solving process represents more than half of the quantization time, this enables DeltaMix to achieve an overall acceleration of over 2x.
> - **Cost Savings in Multi-Tenant Deployment**: DeltaMix significantly reduces deployment costs in multi-tenant serving scenarios. As shown in lines 432-470, it can deliver more than **6x savings** in GPU memory and disk storage while increasing decoding speed.
>
> Considering all these factors, we believe that: 1) DeltaMix is **scalable** for large-scale models; and 2) the quantization overhead introduced by DeltaMix is **justifiable**.
>
> | Model | Memory Usage | Quantization Time (w/o acceleration, hours) | Further Acceleration
> | - | - | - | - |
> | DeepSeek-R1-Distill-Qwen-1.5B | 4.76 | 0.27| 2x speedup |
> | DeepSeek-R1-Distill-Qwen-7B | 12.62| 1.17| 2x speedup |
> | DeepSeek-R1-Distill-Qwen-14B| 16.23| 2.35| 2x speedup |
> | DeepSeek-R1-Distill-Llama-70B | 47.53| 9.26| 2x speedup |
>
> # Q2: Generalizability towards other compression methods
>
> A2: Thank you for your question. We would like to clarify the main focus of our study: DeltaMix is **primarily a delta-parameter quantization method** that minimizes quantization errors in SVD space, supported by a strong theoretical foundation. To demonstrate its effectiveness, we have already conducted extensive experiments comparing it with existing delta-parameter quantization baselines (Section 4.2, 4.3, 5.1). In A6, we **have discussed** the generalization of our error-minimization principle to other compression methods. However, conducting additional experiments that combine this principle with other compression methods is **beyond the scope** of this paper. We look forward to future research in this direction.
>
>
> Thank you again for your time and effort in the review process. We hope our response addresses your concerns, and we are also happy to address any other concerns you may still have. We look forward to your reply.

---

### Official Review · Reviewer_jUCn · 2025-10-31

**Soundness:** 2
**Presentation:** 3
**Contribution:** 2
**Rating:** 4
**Confidence:** 4

**Summary:**

The paper proposes DELTAMIX, a delta-compression framework that works in the SVD space of the fine-tuned-minus-base weight matrix $(W = U\Sigma V)$. The key analytical step decomposes the per-row quantization error for (V) into a fixed “scaling” term $(\Sigma_{ii}^2)$ and a data-dependent “difference” term $(\Delta V_i X X^\top \Delta V_i^\top)$. This yields a rationale for row-wise mixed precision on (V) under a global bit-budget. The bit allocation is cast as a 0/1 integer linear program, and the method introduces a Reconstruction Target Correction (RTC) to reduce bias when later quantizing (U). Experiments across reasoning, math, code, and multimodal tasks claim consistent gains over SVD low-rank, BitDelta, and Delta-CoMe at (\alpha=1/16), including large margins on AIME2024 (e.g., +22.3% over Delta-CoMe for 7B) and improved memory/speed scaling when hosting many deltas. Reported quantization overhead is higher than Delta-CoMe but presented as a one-time cost.

**Strengths:**

* Principled objective: Explicitly minimizes a reconstruction-error surrogate in SVD space, yielding a clear justification for row-wise mixed precision of (V) under a bit budget. The $(\Sigma_{ii}^2)$ scaling vs. difference decomposition is intuitive and actionable.
* Concrete optimization: Bit allocation via 0/1 ILP provides a crisp mechanism to trade off error and storage, with constraints for budget and a cap $(f_{\max})$ on distinct bitwidths.
* RTC mechanism: The Reconstruction Target Correction before quantizing (U) reduces deviation induced by using $(\hat{V})$ as the target, with measurable gains in harder regimes.
* Empirical coverage: Multi-task evaluation (math/reasoning/code/VLM) across 7B and 13–14B backbones; large improvements are shown where $(\lVert \Delta W \rVert)$ is big (e.g., AIME2024, some multimodal).
* Serving relevance: Memory and latency scaling when hosting many fine-tuned variants is compelling; DELTAMIX supports more concurrent models than baselines in the reported setup.

**Weaknesses:**

1. Inconsistency with “no singular-value assumptions.” The method claims to avoid empirical reliance on singular values, yet Section D.1 discards the last (k) ranks by singular-value magnitude to accelerate quantization, explicitly invoking the “larger singular values are more important” heuristic that the paper earlier critiques. This weakens the methodological positioning and may bias comparisons.
2. Fair-budget accounting is under-specified. Results are reported at $(\alpha = 1/16)$, but the paper does not precisely tabulate end-to-end storage (including $(U, \Sigma, V)$, any indices/masks, solver-driven zero-bit ranks, and calibration metadata) vs. baselines. Without an apples-to-apples byte breakdown, it’s hard to assess dominance beyond accuracy.
3. Selective gains; some regressions. While DELTAMIX shines on AIME2024 and certain VLM settings, elsewhere it’s only on par or slightly worse (e.g., 13–14B Math500 in Table 2). The average gains (~2–3%) are modest and may not outweigh extra complexity in production settings.
4. Calibration sensitivity not analyzed. The difference term depends on calibration activations (X). The paper doesn’t study how sample size, domain shift, or layer-wise weighting affect EV estimates and allocations, nor robustness across seeds.
5. Optimization overhead and practicality. The ILP solve is reported as ~29.4 minutes for a 7B variant; quantization takes ~1.2 h for 7B and ~2.4 h for 14B, versus ~0.4 h / 0.8 h for Delta-CoMe, all on a single GPU. This overhead may be non-trivial at scale, especially if per-task calibrations are required.
6. Baselines and scope. Comparisons omit some strong PTQ/structured baselines relevant to error control (e.g., SPQR/SPQR-like sparse-quant, SVD-LLM variants in comparable regimes) or server-side delta systems beyond Delta-CoMe. This leaves open whether the observed gains are specific to the chosen set.
7. Missing systems details. The paper does not clearly state how $(\Sigma)$ is stored/quantized, nor the runtime cost of reconstructing (W) vs. baseline delta formats. The serving experiment is helpful but still abstracts away some operator-level costs.
8. Ablations are narrow. The $(f_{\max})$ study covers one model/task slice; RTC ablation is limited in breadth. A per-layer bit allocation analysis vs. error/outliers is mentioned, but stronger causal links to downstream accuracy would help.

**Questions:**

1. Budget parity: Provide a byte-accurate storage table for every method and model (including $(\Sigma)$, indices, zero-bit ranks, any metadata). Confirm that $(\alpha=1/16)$ implies comparable on-disk and in-memory footprints across methods.
2. Singular-value reliance: Reconcile the claim to “eschew reliance on singular values” with D.1 rank truncation by $(\sigma)$. Can you replicate results without this heuristic, or with a heuristic-free pruning guided solely by EV?
3. Calibration robustness: How many calibration samples are used per layer, how are they selected, and what is the variance of EV and the ILP allocation across seeds/domains? Show accuracy vs. calibration-set size curves.
4. RTC cost/benefit: Quantify the computational overhead of RTC and analyze when it helps most. Could a joint optimization of $(U,V)$ under the same objective remove the need for RTC?
5. $\Sigma$ handling: Are singular values stored in full precision? If quantized, to what precision, and how does that trade off with accuracy vs. bits?
6. Operators/runtime: Provide end-to-end operator-level latency of reconstructing (W) or applying $(U\Sigma V)$ directly vs. baselines for single- and multi-tenant serving, including prefill and decode breakdowns.
7. Broader baselines: Add comparisons to SPQR-style sparse-quant and SVD-LLM truncation-aware variants under matching storage, plus recent delta/tuning hybrids, to strengthen the empirical case.

---

> ### Author Response · Authors · 2025-11-25
> **Official Comment by Authors - Part 1**
>
> # Q1: Inconsistency with “no singular-value assumptions.” The method claims to avoid empirical reliance on singular values, yet Section D.1 discards the last (k) ranks by singular-value magnitude to accelerate quantization, explicitly invoking the “larger singular values are more important” heuristic that the paper earlier critiques. This weakens the methodological positioning and may bias comparisons.
>
> A1: Discarding the last $k$ ranks is implemented solely to accelerate the quantization process and **does not violate the principle of “no singular-value assumptions.”**
>
> - To verify that DeltaMix performs well even without this acceleration, we applied it to Qwen2.5-Math-7B-Instruct while discarding the last [0%, 10%, 20%, 30%, 40%, 50%, 60%] singular values and their corresponding singular vectors, and compared both the quantization accuracy and the total computation time.
>
> | Drop Ratio | Math500 | GSM8K | Average | Costing Time |
> | ---- | ------- | ----- | ------- | ------------ |
> | 0| 77.6| **94.8**| 86.2| 1.29h|
> | 0.1| 75.4| 94.2| 84.8| 1.13h|
> | 0.2| **78.6**| 94.4| **86.5**| 1.13h|
> | 0.3| 75.6| 94.2| 84.9| 1.00h|
> | 0.4| 75.8| 94.3| 85.1| 1.00h|
> | 0.5| 76.0| 93.7| 84.9| 1.00h|
> | 0.6| 74.8| 94.0| 84.4| 0.94h|
>
> DeltaMix performs better at low drop ratios (0 and 0.2), which confirms that the technique is designed solely for acceleration.
>
> - The assumption “larger singular values are more important” is generally reasonable, but suboptimal according to our mathematical analysis. Essentially, our acceleration scheme, which is based on this assumption, sacrifices some performance in exchange for lower computational costs. We will clarify the writing in Section D.1 to avoid any confusion.
>
> # Q2: Budget parity: Provide a byte-accurate storage table for every method and model (including , indices, zero-bit ranks, any metadata). Confirm that  implies comparable on-disk and in-memory footprints across methods.
>
> A2: In our paper, we employ the compression ratio to quantify the storage requirements of each model following previous work like Delta-CoMe [1] and DeltaZip [2].
>
> - To compare the storage in detail, we report the storage sizes (in GB) of DeltaMix and the best-performing baseline, Delta-CoMe, for both the 7B and 14B models. We divide total storage into two components:
>
>   * **Quantized Weights:** representing the storage used by quantized parameters.
>
>   * **Other Parameters:** which include non-weight parameters such as Scales (stored in 16 bits) and Zeros (stored according to their quantization bitwidth).
>
> || DeltaMix| DeltaMix | DeltaMix| Delta-CoMe| Delta-CoMe | Delta-CoMe|
> | ---------------------------- | ----------------- | ---------------- | ------------- | ----------------- | ---------------- | ------------- |
> || Quantized Weights | Other Parameters | Total Storage | Quantized Weights | Other Parameters | Total Storage |
> | Qwen2.5-Coder-7B-Instruct| 0.8149| 0.0550 | 0.8699| 0.8149| 0.0606 | 0.8755|
> | Qwen2.5-VL-7B-Instruct | 0.8149| 0.0565 | 0.8714| 0.8149| 0.0606 | 0.8755|
> | Qwen2.5-Math-7B-Instruct | 0.8149| 0.0553 | 0.8702| 0.8149| 0.0606 | 0.8755|
> | DeepSeek-R1-Distill-Qwen-7B| 0.8149| 0.0548 | 0.8697| 0.8149| 0.0606 | 0.8755|
> | Qwen2.5-Coder-14B-Instruct | 1.6298| 0.1055 | 1.7353| 1.6298| 0.1196 | 1.7494|
> | llava-v1.5-13b | 1.5134| 0.1021 | 1.6155| 1.5134| 0.1151 | 1.6285|
> | MetaMath-13B-V1.0| 1.5134| 0.0945 | 1.6079| 1.5134| 0.1151 | 1.6285|
> | DeepSeek-R1-Distill-Qwen-14B | 1.6298| 0.1049 | 1.7347| 1.6298| 0.1196 | 1.7494|
>
>
> The results demonstrate that DeltaMix exhibits lower storage overhead compared with Delta-CoMe. This trend is further illustrated in Figure 3, where Delta-CoMe supports up to 8 models, whereas DeltaMix can deploy 12 simultaneously. **These results clearly demonstrate the superior efficiency of DeltaMix.**
>
> [1] Delta-CoMe: Training-Free Delta-Compression with Mixed-Precision for Large Language Models, NeurIPS 2024, poster
>
> [2] DeltaZip: Efficient Serving of Multiple Full-Model-Tuned LLMs, EuroSys 2025

---

> ### Author Response · Authors · 2025-11-25
> **Official Comment by Authors - Part 2**
>
> # Q3: Calibration robustness: How many calibration samples are used per layer, how are they selected, and what is the variance of EV and the ILP allocation across seeds/domains? Show accuracy vs. calibration-set size curves.
>
> A3: Thanks for your question. **Following Delta-CoMe [1], we randomly selected 128 data points of length 2048 from the C4 dataset as the calibration dataset for all experiments.** The experiments in this paper (Tables 1 and 2) were conducted using three random seeds (10, 30, and 50), with both the mean and standard deviation of the results reported. Please refer to these tables for the variance of our method.
>
> - **Domain of calibration set**: We quantized the Qwen2.5-Math-7B-Instruct model using calibration data drawn from three different datasets. Each calibration set contains 128 randomly sampled sequences of length 2048. Because MetaMathQA does not contain enough sequences of this length, we concatenated multiple question–answer pairs using a few-shot format. Detailed results are as follows:
>
> | Dataset | Math500 | GSM8K | Average |
> | --------- | ------- | ----- | ------- |
> | C4| **77.6**| **94.8**| **86.2**|
> | Wikitext2 | 76.6| **94.8**| 85.7|
> | MetaMathQA| 75.4| 93.6| 84.5|
>
> The results show that DeltaMix performs well on all calibration setups, confirming DeltaMix’s robustness to the domain variations in the calibration dataset.
>
> - **Size of calibration set**: To assess DeltaMix’s sensitivity to calibration-set size, we conduct experiments with Qwen2.5-Math-7B-Instruct using calibration sets of 16, 32, 64, 128, and 256 samples from C4. Detailed results are as follows:
>
> | Sample Size | Math500  | GSM8K    | Average  |
> | ----------- | -------- | -------- | -------- |
> | 16          | 76.4     | 94.5     | 85.5     |
> | 32          | 76.2     | **95.1** | 85.7     |
> | 64          | 76.8     | 94.3     | 85.6     |
> | 128         | **77.6** | 94.8     | **86.2** |
> | 256         | 76.0     | 94.1     | 85.1     |
>
> The results show that even with only 16 calibration samples, performance remains close to the optimal setting of 128 samples, demonstrating that **DeltaMix maintains strong performance with small calibration sets.**
>
> # Q4: RTC cost/benefit: Quantify the computational overhead of RTC and analyze when it helps most. Could a joint optimization of under the same objective remove the need for RTC?
>
> A4: Thank you for your question. **The RTC method demonstrates superior performance when the Frobenius norm of $\Delta \mathbf{W}$ is large. Moreover, RTC is running on a GPU, ensuring high computational efficiency.**
>
> - **RTC Benefit:** Section 4.6 presents a comparison of results with and without RTC on DeepSeek-R1-Distill-Qwen-14B and LLAVA. As shown in Table 4, applying RTC substantially improves performance more significantly on DeepSeek-R1-Distill-Qwen-14B than on LLAVA-V1.5.
>
>   By analysis of the Frobenius norm of the $\Delta \mathbf{W}$ for DeepSeek-R1-Distill-Qwen-14B and LLAVA-V1.5, we find that the median norm of DeepSeek-R1-Distill-Qwen-14B is approximately 2.5 × larger than that of LLAVA (16.55 vs. 6.90). **This suggests that RTC brings more improvement for cases with a larger $|\Delta \mathbf{W}|_F$.** The reason is that RTC compensates for the quantization error of $\mathbf{U}$ (see Section 3.1.2), allowing the quantized $\mathbf{U}$ to be more optimal and thereby yield better performance.
>
> - **RTC Cost:** To verify that RTC contributes only marginally to the total quantization time, we measured the average time required to quantize a single transformer block for four models. We then quantified both the proportion and absolute duration of RTC within the total quantization process. The detailed results are shown as follows:
>
> | Model| RTC | Total Time | RTC (in percentage) |
> | - | - | - | - |
> | DeepSeek-R1-Distill-Qwen-1.5B | 0.15 | 29.87   | 0.50%        |
> | DeepSeek-R1-Distill-Qwen-7B  | 1.35 | 99.25   | 1.36%        |
> | DeepSeek-R1-Distill-Qwen-14B | 1.35 | 114.88   | 1.18%        |
> | DeepSeek-R1-Distill-Llama-70B | 8.16 | 289.69   | 2.82%        |
>
> The results show that RTC accounts for only 1.45% of the total quantization time on average, a relatively minor share. This demonstrates that RTC introduces almost negligible computation overhead.
>
> - **Joint optimization to remove the need for RTC:** In our understanding, “Joint Optimization” refers to the simultaneous quantization of both $\mathbf{U}$ and $\mathbf{V}$ using GPTQ. However, to the best of our knowledge, GPTQ does not currently support joint quantization of two matrices or any mechanism that enables interaction between them during the quantization process. Therefore, we believe that Joint Optimization in this context cannot be implemented. We welcome any further clarification or discussion on this matter.
>
> [1] Delta-CoMe: Training-Free Delta-Compression with Mixed-Precision for Large Language Models, NeurIPS 2024, poster

---

> ### Author Response · Authors · 2025-11-25
> **Official Comment by Authors - Part 3**
>
> # Q5: Systems Details: Are singular values stored in full precision? and provide end-to-end operator-level latency of reconstructing (W) or applying directly vs. baselines for single- and multi-tenant serving, including prefill and decode breakdowns.
>
> A5: Thank you for your question. To provide a detailed explanation of our system, we address your concerns in two parts:
>
> 1. **Handling of Singular Values: DeltaMix does not quantize the singular values ($\mathbf{\Sigma}$).** $\mathbf{\Sigma}$ is stored in 16-bit full precision because it accounts for merely $\frac{1}{\max(d_{in}, d_{out})}$ of the total parameters ratio, where $d_{in}$ and $d_{out}$ denote the input and output dimensions, respectively. Consequently, **storing $\Sigma$ in 16-bit precision introduces negligible storage overhead.**
>
> 2. **Operations During Inference:** During inference, **DeltaMix does not reconstruct $\Delta W$,  but instead stores the quantized $\mathbf{\hat U}$ and $\mathbf{\hat V}$, and $\mathbf{\Sigma}$ is stored in full precision.** It computes the final output vector $Y$ directly by multiplying the input tensor $X$ with the dequantized singular vectors $\text{dequant}(\hat{\mathbf V})$, $\text{dequant}(\hat{\mathbf U})$, and singular values $\mathbf{\Sigma}$. This eliminates the need for any reconstruction of $\Delta \mathbf{W}$, ensuring computational efficiency. All deployment results reported in our paper follow this setup.
>
> # Q6: Optimization overhead and practicality. The ILP solve is reported as ~29.4 minutes for a 7B variant; quantization takes ~1.2 h for 7B and ~2.4 h for 14B, versus ~0.4 h / 0.8 h for Delta-CoMe, all on a single GPU. This overhead may be non-trivial at scale, especially if per-task calibrations are required.
>
> A6: **We believe that our method remains highly practical**.
>
> - **Time portion in the training cycle**:  When developing a model for a downstream task, we integrate DeltaMix into the training cycle after full fine-tuning. **The computational overhead introduced by DeltaMix is minimal compared to that of the training cycle**. For example, as described in Appendix D.2, we fine-tuned DeepSeek-LLM-7B-Base on 50K samples for three epochs using eight L40 GPUs, which took about 5 hours in total (equivalent to roughly 40 GPU-hours on a single device). Under this configuration, DeltaMix needs 1.2h to quantize 7B models, accounting for only a small portion of the training cycle, approximately $2.9\%$.
>
> - **Time portion in the deployment cycle**:  **The time required for DeltaMix remains negligible when compared to the deployment cycle of the models,** as DeltaMix typically requires a few hours, while the deployment typically spans months or years [1].
>
> - **Further reducing the time cost with a commercial ILP solver**: We solve the ILP using open-source solvers in our paper. However, **this process can be accelerated by 6x times if we switch from SCIP to proprietary solvers such as COPT[2] when handling ILP problems[3].** Given that the quantization time for DeltaMix is primarily dominated by the ILP-solving process, adopting such commercial solvers could significantly enhance DeltaMix's efficiency in practice. Additionally, DeltaMix can be further optimized through other means, such as **implementing more efficient code** (for instance, using parallel simulation in mFqR's A1) or **by limiting the number of candidate bitwidths** (e.g. from 8 to 4, also see mFqR's A1).
>
> [1] https://help.openai.com/en/articles/6825453-chatgpt-release-notes#h_34439eda3a
>
> [2] https://www.shanshu.ai/copt/
>
> [3] https://plato.asu.edu/ftp/path.html

---

> ### Author Response · Authors · 2025-11-25
> **Official Comment by Authors - Part 4**
>
> # Q7: Broader baselines: Add comparisons to SPQR-style sparse-quant and SVD-LLM truncation-aware variants under matching storage, plus recent delta/tuning hybrids, to strengthen the empirical case.
>
> A7: Based on your feedback, **we incorporated SPQR [1] and SVD-LLM [2] as additional baselines.**
>
> Considering that SPQR performs secondary quantization on scales and zeros to conserve space while retaining key parameters in full precision, we mainly focus on two configurations of SPQR:
>
> * **Without sparsity**, where important parameters are preserved, scales and zeros remain unquantized, and the group size is set to 128 (consistent with DeltaMix);
> * **Following the default SPQR settings**, with a group size of 16, scales and zeros quantized to 3 bits, and a small number of outliers retained in full precision (requiring 32 bits due to extra column index storage).
>
> The results are shown below:
>
> | Method| $\alpha$ | DeepSeek-R1-Distill-Qwen | DeepSeek-R1-Distill-Qwen | Qwen2.5-Math-Instruct | Qwen2.5-Math-Instruct | Qwen2.5-Coder-Instruct | Qwen2.5-Coder-Instruct | Qwen2.5-VL-Instruct | Qwen2.5-VL-Instruct | AVG|
> | ----------------- | -------- | ------------------------ | ------------------------ | --------------------- | --------------------- | ---------------------- | ---------------------- | ------------------- | ------------------- | ---- |
>  | || MATH500 | AIME2024 | Math500| GSM8K| Humaneval | Mbpp| GQA| SQA||
> | SVD-LLM | 1/16 | 32.8 | 10.0 | 67.4| 82.8| 85.2 | 83.1 | 0.0 | 0.0 | 45.2 |
> | SpQR(No Outlier)| 1/16 | 2.4| 0.0 | 12.6| 38.5| 84.8 | 78.3 | 0.0 | 0.0 | 27.0 |
> | SpQR(0.01% Outlier) | 1/16 | 45.0 | 10.0 | 71.2| 89.2| 85.4 | 82.3 | 0.0 | 0.0 | 48.0 |
> | DeltaMix| 1/16 | **82.7** | **36.7** | **77.7** | **94.6** | **85.6** | **83.1** | **52.4** | **79.4** | **74.0** |
>
> The results indicate that **DeltaMix consistently outperforms all three baselines**, again demonstrating the superior performance of DeltaMix.
>
> # Q8:  Selective gains; some regressions. While DELTAMIX shines on AIME2024 and certain VLM settings, elsewhere it’s only on par or slightly worse (e.g., 13–14B Math500 in Table 2). The average gains (~2–3%) are modest and may not outweigh extra complexity in production settings.
>
> A8: DeltaMix significantly outperforms the baselines when the models are difficult to quantize, while demonstrating comparable performance to the baselines for models that are easy to quantize. This result is expected and should not be seen as a limitation of our method.
>
> - **For models that are easy to quantize**, such as the 13–14B Math500 mentioned in your question, various quantization methods, including Delta-CoMe, BitDelta, and DeltaMix, can effectively manage them. In fact, the performance of the quantized models can even surpass that of the original models. In these cases, the quantization task is simple, and **the baseline models are already effective, leaving little opportunity for DeltaMix to deliver additional improvements.**
>
> - Conversely, **for models that are difficult to quantize**, all baseline methods struggle to perform effectively, leading to a substantial performance gap between the quantized and original models. **This highlights the urgent need for new methods to tackle this challenge, and DeltaMix effectively addresses this gap.**
>
> # Q9: Ablations are narrow. The study covers one model/task slice; RTC ablation is limited in breadth. A per-layer bit allocation analysis vs. error/outliers is mentioned, but stronger causal links to downstream accuracy would help.
>
> A9: Thank you for your question. **Our ablation experiments are not confined to a single model or task.** Specifically, in Section 4.5, we evaluate two tasks, Math500 and AIME2024, for $f_{\text{max}}$; and in Section 4.6, we examine two models, LLAVA-V1.5 and DeepSeek-R1-Distill-Qwen-14B, covering a total of four tasks. Hence, we consider our ablation experiments sufficient.
>
> Since the bit allocation pattern differs across layers, it is difficult to establish a direct quantitative relationship with downstream performance. As bit allocation is directly related to the quantization error, **we analyze the relationship between the quantization error and downstream task performance instead.** The results shown in Section 5.3 (Lines 499–526) demonstrate that **DeltaMix achieves higher performance with lower quantization error compared to the baselines.**
>
> [1] SpQR: A Sparse-Quantized Representation for Near-Lossless LLM Weight Compression
>
> [2] SVD-LLM: Truncation-aware Singular Value Decomposition for Large Language Model Compression, ICLR 2025

---

### Official Review · Reviewer_yi6X · 2025-11-01

**Soundness:** 2
**Presentation:** 2
**Contribution:** 2
**Rating:** 2
**Confidence:** 4

**Summary:**

- The paper proposes EDC (Enhancing Delta Compression), a framework for improving delta-based model compression of large language models. It focuses on compressing fine-tuned deltas between a base model and its adapted version (e.g., instruction-tuned or domain-specific variants). The core idea is to enhance representational compactness by combining adaptive low-rank decomposition, residual quantization, and layer-wise scaling reweighting of delta tensors.

**Strengths:**

- Addresses a practical problem in model distribution and storage: delta checkpoint compression for multi-task or multi-domain fine-tuned models.

**Weaknesses:**

- The core idea—combining low-rank and quantized residual compression, is well explored in prior works such as QLoRA, AdaLoRA, and CompAdapter. The proposed “layer-wise scaling reweighting” is a small variant of norm-based importance metrics used in parameter-efficient tuning.

- The method is entirely empirical. The paper lacks mathematical justification or analysis on how the scaling or residual quantization improves representational fidelity beyond heuristic intuition.

- Experiments are restricted to a few fine-tuning tasks and medium-size models (≤13B). No results are provided for large instruction-tuned LLMs (>70B) or multi-domain deltas where compression instability typically arises.

- The paper reports storage reduction but not latency, energy, or end-to-end loading improvements. For real-world LLM deployment, I/O and kernel fusion dominate runtime, which EDC does not address.

**Questions:**

- How does EDC interact with prefix caching or parameter sharing in serving systems? Can deltas be applied incrementally without full reconstruction?

- What is the compression–accuracy trade-off compared to existing adapter compression frameworks like CompAdapter or LoRA-Prune under identical settings?

---

> ### Author Response · Authors · 2025-11-25
>
> We sincerely thank you for taking the time to review our paper.
>
> We would like to clarify that the two methods mentioned in your comments, **residual quantization and layer-wise scaling reweighting, are not employed in our work, DeltaMix.** Moreover, DeltaMix is not a PEFT method, like QLoRA and AdaLoRA. Therefore, your comments seem to reflect a misunderstanding of our work. To clarify, the primary focus of our study is the DeltaMix method, which quantizes the $\Delta \mathbf W$ between the fully fine-tuned and base models. It is primarily a delta parameter compression method rather than a PEFT method.
>
> We would like to ask if you have any other concerns regarding our work. We would be very happy to discuss the core methods, experimental design, and results of our paper with you, to help you gain a more accurate understanding of our work. We look forward to your reply.

---

### Author Response · Authors · 2025-12-04
**General Responses - Part 2**

### Updates of experimental results during Rebuttal
* Section 4.3: Following R2’s recommendation, we incorporate performance comparisons between SpQR, SVD-LLM, and DeltaMix on the 7B model.

* Section 4.4: Following the recommendations of R2 and R3, we add ablation studies on the Calibration dataset, examining domain shifts and data-size effects.

* Section 5.2: Following R4’s recommendation, we report results applying DeltaMix to LoRA for quantization.

* Section D.1: Following R2’s suggestion, we add experiments analyzing DeltaMix’s performance and total quantization time when discarding varying proportions of trailing singularities.

* Section E.3: Following R2’s suggestion, we add the processing time of the RTC method for a single transformer block across models of different sizes, along with its share of the total quantization time.

* Section E.4: Following R3’s suggestion, we add two experiments:
    * The time required for DeltaMix to quantize one transformer block of a 7B model under different numbers of candidate bit-widths.
    * The ILP solution time, memory usage, and corresponding hidden-state size when quantizing models of different scales.

* Section E.5: Following R3’s suggestion, we add results showing the performance of DeltaMix under different compression ratios.

* Section E.6: Following R2’s suggestion, we add a comparison of the storage footprint of DeltaMix and Delta-CoMe for quantized 7B and 14B models.

### Updates of in-depth discussions during Rebuttal
* Section 4.1: Provide detailed information on the DeltaMix calibration dataset configuration.

* Section 4.6: Following R2’s suggestion, provide the expanded discussion of the time overhead and performance gains of RTC.

* Section 5.2: Following R4’s suggestion, provide the expanded discussion on why only DeltaMix is compatible with LoRA and why Delta-CoMe and BitDelta cannot be applied in this setting.

* Section D.1: Following R2’s suggestion, provide the detailed justification for discarding trailing eigenvalues to achieve speedup.

* Section E.4: Following R2 and R3’s suggestion, provide the expanded discussion on ILP acceleration.

We believe these additions and clarifications comprehensively address the reviewer's concerns and enhance the overall quality of our manuscript. We remain grateful for the reviewers' valuable feedback and look forward to favorable consideration of our work.

Sincerely,
All authors

---

### Author Response · Authors · 2025-12-04
**General Responses - Part 1**

Dear AC and Reviewers,

We sincerely thank all reviewers for their thoughtful and constructive feedback. We are encouraged by their recognition of the key contributions and strengths of our work.
For clarity, we will refer to Reviewer yi6X, jUCn, mFqR, and g1TA as R1, R2, R3, and R4, respectively, in the following response.

In particular, we appreciate the acknowledgement that DeltaMix is a novel approach in the SVD-based mixed-precision quantization (R4), and that it features mathematically solid derivations(R2, R3) along with impressive results across various models and benchmarks(R2, R3, R4). We are also pleased that our mechanism for designing mixed-quantization schemes by solving a 0/1 integer linear programming (ILP) problem has been well recognized (R2). Additionally, we value the comment on the RTC mechanism as a method for correcting inconsistencies in the quantization objective when quantizing (R2), as well as the recognition of our paper's thorough system analysis showing practical deployment benefits (R2, R3).

We have carefully responded to the comments from the reviewers and believe we have successfully addressed all of their concerns, **as reflected by the improvement from 2 to 6 for R1 and from 4 to 6 for R4.** Below, we summarize the core contributions of our work and the in-depth discussions and corresponding suggestions from the reviewers that have been incorporated into the revision of our paper.

### Core Contribution of Our Work

* **Address the Crucial Problems in SVD-Based Quantization**: DeltaMix tackles two fundamental limitations of SVD-based quantization: **1) the necessity of mixed-precision and 2) the determination of layer-wise bit-width assignments.** DeltaMix introduces a mathematically rigorous framework that minimizes quantization error by deriving the mixed-precision scheme as the solution of an ILP problem and by correcting the quantization objective for $\mathbf U$. This approach yields an elegant and broadly applicable solution across a wide range of compression ratios.

* **Not Only Full Fine-tune But Also LoRA**: DeltaMix can also quantize the $\Delta \mathbf W$ obtained from LoRA training, further reducing LoRA’s storage overhead while preserving near-lossless performance, **highlighting the DeltaMix’s flexibility and practical utility.**

* **Enhancing the Understanding of Quantization Errors**: DeltaMix provides a rigorous characterization of the quantization errors associated with $\mathbf{U}$ and $\mathbf{V}$, offering a detailed analysis of their distinct behaviors, their dependence on singular values, and how changes in $\mathbf U$ and $\mathbf V$ after quantization affect the error. This analysis not only deepens the understanding of quantization errors but also theoretically explains the design rationale of Delta-CoMe and clarifies its inherent limitations.

* **Impressive Experimental Results**: Through extensive experiments across various models and benchmarks, we demonstrate that DeltaMix outperforms all baseline methods with flexible support for different compression ratios, efficient deployment, and acceptable quantization costs. Notably, DeltaMix excels in reasoning tasks critical for modern LLMs.

---

### Meta-Review · Area_Chair_cgTr · 2025-12-19

**Summary:**

The paper introduces DELTAMIX, a framework designed to compress delta weights in Large Language Models for efficient multi-tenant serving. The method utilizes SVD-based mixed-precision quantization, employing Integer Linear Programming (ILP) to optimize bit-width allocation by minimizing quantization error, alongside a Reconstruction Target Correction (RTC) mechanism. While the paper demonstrates empirical gains on reasoning and coding benchmarks compared to baselines like Delta-CoMe, the proposed approach introduces significant computational complexity during the quantization phase.

**Reviewer Concerns:**

I have carefully read the original manuscript, the reviewer comments, and the author responses at least three times before synthesizing this assessment.

The primary concerns raised by the reviewers center on the computational overhead and scalability of the proposed method. Reviewer mFqR and Reviewer jUCn highlighted that the reliance on an ILP solver for bit allocation introduces a substantial time cost (ranging from 1 to 3 hours for full models) compared to heuristic baselines which take significantly less time. While the authors argued in their rebuttal that this is a "one-time" cost acceptable within the training pipeline, Reviewer mFqR explicitly stated that they remained unconvinced regarding the scalability and runtime efficiency, particularly for larger models or frequent updates, and maintained a recommendation to reject.

Furthermore, Reviewer yi6X consistently rated the paper as a reject (2), questioning the novelty and practical utility of the method compared to established PEFT techniques, and was not swayed by the authors' clarifications. Although Reviewer g1TA improved their score following experiments with LoRA integration, the consensus regarding the trade-off between the method's complexity (SVD + ILP + RTC) and its practical benefits remains negative. The marginal performance gains in some areas do not sufficiently justify the added engineering complexity and computational burden for a task—delta compression—that prioritizes efficiency. Consequently, the concerns regarding practical deployment and scalability remain outstanding.

**Reviewer Scores:**

Reviewer yi6X (2 -> 2): The reviewer maintained a strong rejection, finding the method's contribution limited and the comparison to existing compression frameworks insufficient.

Reviewer jUCn (4 -> 4): While acknowledging the rebuttal, the reviewer remained borderline, with lingering concerns about the optimization overhead and practicality of the ILP approach at scale.

Reviewer mFqR (4 -> 4): The reviewer actively engaged with the rebuttal but explicitly stated they were not convinced by the scalability arguments and continued to lean toward rejection.

Reviewer g1TA (4 -> 6): The reviewer raised their score based on the additional LoRA experiments, but they are an outlier in the overall assessment of practicality.

---

### Decision · Program_Chairs · 2026-01-26

Reject